# ARID1A mutations confer intrinsic and acquired resistance to cetuximab treatment in colorectal cancer

Radia M. Johnson [1] ✉, Xueping Qu [2] ✉, Chu-Fang Lin[3], Ling-Yuh Huw[2], Avinashnarayan Venkatanarayan[4], Ethan Sokol[5], Fang-Shu Ou [6], Nnamdi Ihuegbu[7], Oliver A. Zill [1], Omar Kabbarah[2], Lisa Wang[3], Richard Bourgon [1], Felipe de Sousa e Melo [4], Chris Bolen[1], Anneleen Daemen[1], Alan P. Venook[8], Federico Innocenti[9], Heinz-Josef Lenz [10] & Carlos Bais [2] ✉

Most colorectal (CRC) tumors are dependent on EGFR/KRAS/BRAF/MAPK signaling activation. ARID1A is an epigenetic regulator mutated in approximately 5% of non-hypermutated CRC tumors. Here we show that anti-EGFR but not anti-VEGF treatment enriches for emerging ARID1A mutations in CRC patients. In addition, we find that patients with ARID1A mutations, at baseline, are associated with worse outcome when treated with cetuximab- but not bevacizumab-containing therapies; thus, this suggests that ARID1A mutations may provide both an acquired and intrinsic mechanism of resistance to anti-EGFR therapies. We find that, ARID1A and EGFR-pathway genetic alterations are mutually exclusive across lung and colorectal cancers, further supporting a functional connection between these pathways. Our results not only suggest that ARID1A could be potentially used as a predictive biomarker for cetuximab treatment decisions but also provide a rationale for exploring therapeutic MAPK inhibition in an unexpected but genetically defined segment of CRC patients.

Advances in the treatment of metastatic colorectal cancer (mCRC) with targeted therapies, including anti-EGFR (e.g., cetuximab) and anti-VEGF (e.g., bevacizumab) therapy in combination with chemotherapy, have improved the outcome for CRC patients[1]. Alterations in genes involved in EGFR/MAPK signaling play an important role in primary (intrinsic) and secondary (acquired) resistance to anti-EGFR therapy in RAS/BRAF wild-type (WT) patients as well as combination therapy with BRAF inhibitors in BRAFV600E patients[2,3]. Known mechanisms of resistance to anti-EGFR frequently include upstream mutations in the extracellular domain of EGFR that directly confer resistance to antibody blockade[4] and also EGFR downstream

signaling reactivation mainly through KRAS, NRAS, and BRAF, activating mutations[5–7], collectively referred to here as extended RAS (eRAS)/BRAF mutations. Additionally, less frequent genetic mechanisms of resistance to cetuximab likely through sustained ERK signaling have been identified, including MAP2K1[8–10] and NF1[11,12] mutations and KRAS, MET, and ERBB2 amplification[13–15] among others. Yet, for a number of additional patients, resistance is readily observed but the underlying mechanisms driving it have remained poorly defined.

By leveraging the genomic biomarker data collected from consenting patients who participated in the Cancer and Leukemia Group B

[1]Bioinformatics & Computational Biology, Genentech, Inc., South San Francisco, CA, USA. [2]Oncology Biomarker Development, Genentech, Inc., South San Francisco, CA, USA. [3]Real World Data Science Analytics, Genentech, Inc., South San Francisco, CA, USA. [4]Discovery Oncology, Genentech, Inc., South San Francisco, CA, USA. [5]Cancer Genomics Research, Foundation Medicine, Inc., Cambridge, MA, USA. [6]Alliance Statistics and Data Management Center, Mayo Clinic, Rochester, MN, USA. [7]Guardant Health, Inc, Redwood City, CA, USA. [8]University of California, San Francisco, San Francisco, CA, USA. [9]University of North Carolina at Chapel Hill, Chapel Hill, NC, USA. [10]USC Norris Comprehensive Cancer Center, Los Angeles, CA, USA. ✉e-mail: johnson.radia@gene.com; qu.xueping@gene.com; baiscarlos@gmail.com

(CALGB)/SWOG 80405 trial and other relevant CRC datasets, here we investigate additional mechanisms of resistance to anti-EGFR and anti-VEGF containing therapies. In this work, we show data to support ARID1A mutations confer resistance to cetuximab treatment in colorectal cancer.

## Results

### Longitudinal cfDNA analysis of first-line (1L) metastatic CRC uncovered selected enrichment of ARID1A mutations in patients treated with cetuximab but not bevacizumab

To investigate treatment-specific mutations that underlie resistance to treatment with chemotherapy + bevacizumab or cetuximab in 1L mCRC, we performed targeted sequencing of circulating free DNA (cfDNA) from available longitudinal plasma samples collected at baseline (PRE, $n = 354$), during treatment (OTH, $n = 254$), and/or at progression or end of the protocol treatment (EOT, $n = 345$) from CALGB/SWOG 80405, a randomized phase III trial in first-line colorectal cancer patients evaluating the efficacy of chemotherapy plus bevacizumab *vs* chemotherapy plus cetuximab treatment. For this study and text clarity, bevacizumab and cetuximab will refer to bevacizumab + chemotherapy and cetuximab + chemotherapy, respectively. Chemotherapy was a patient/physician choice of fluorouracil, leucovorin, and oxaliplatin (FOLFOX) or fluorouracil, leucovorin, and irinotecan (FOLFIRI). The prevalence of mutations detected in the genes measured in cfDNA of bevacizumab- vs. cetuximab-treated patients was similar between both arms at each of the 3 time points evaluated (Supplementary Fig. 1a). Consistent with the literature, we observed reduced circulating tumor DNA (ctDNA) upon chemotherapy treatment, with a subsequent increase at progression[16,17].

To identify treatment-specific induced mutations, we examined the change in relative mutation allele frequency (MAF) from baseline to end of study for 333 patients with samples at both time points. Relative MAF was calculated as the allele frequency of a given mutation relative the highest reported MAF, and was used to account for the variation in the fraction of tumor vs. normal cell DNA in circulation. Non-synonymous single-nucleotide variants (SNVs) with an increase in relative MAF exceeding 25 percent were considered as mutations likely selected during the course of treatment. Of the 18 genes with at least 2 selected mutations detected in either treatment arm (Fig. 1a and Supplemental Data 1), KRAS and ARID1A selected mutations were significantly enriched in patients treated with cetuximab relative to bevacizumab (KRAS adj. $p = 0.004$, odds ratio (OR) = 0.14, 95% confidence interval (CI) [<0.01, 0.6], ARID1A adj. $p = 0.02$, OR = 0.09, 95% CI [<0.01, 0.7], two-tailed Fisher's exact test (Fig. 1b and Supplementary Fig. 1b), while no significant enrichment was observed in bevacizumab- relative to cetuximab-treated patients. Selection of KRAS mutations was expected in cetuximab-treated patients[18,19]. Most selected KRAS mutations had a change in MAF between 25–50%, consistent with its function as a dominant driver mutation, and included recurrent KRAS variants such as G12A/D/V, G13D, A146T (Fig. 1b and Supplementary Fig. 1c, d). Moreover, KRAS selected mutations not previously detected in tissue were observed in cfDNA at both baseline (low MAF) and progression (higher MAF) in 9/10 cetuximab-treated patients. One patient had an acquired KRAS A146T mutation with a concurrent loss of a pre-existing KRAS G12D mutation at progression (Supplementary Fig. 1d), suggesting that most KRAS emerging mutations are present at low frequency in these KRAS WT patients at trial registration/randomization (likely highly subclonal) and are then expanded post-EGFR therapy.

In contrast to KRAS, the enrichment of ARID1A alterations at end of study was unexpected as ARID1A mutations have thus far not been reported as a resistance mechanism to EGFR blockade in CRC patients. For patients with selected ARID1A mutations, the absolute change in MAF exceeded 50 percentage points for 6/7 patients (Fig. 1b), consistent with ARID1A being a tumor suppressor[20,21] and likely requiring

biallelic loss to confer resistance. Among the 6 cetuximab-treated patients with selected ARID1A mutations: two patients (SAARSX and SAANSJ) had a mutation that was already detected (at a lower MAF) at baseline (Fig. 1c and Supplementary Fig. 2); for three patients (SAASGM, SAAAUP, SABCZU) ARID1A mutations were undetected at baseline (Fig. 1d and Supplementary Fig. 2); and the remaining patient (SAANDA) had 2 selected ARID1A mutations (Fig. 1d), with one variant present at baseline (ARID1A exon 20, D1850fs) and the other first detected on treatment (ARID1A Q309*). In contrast, the only bevacizumab-treated patient (SAAAKR) with a selected ARID1A mutation had an ARID1A exon 7 Q758fs, exon 20 G2069fs, and exon 20 I907fs alteration detected at baseline with only the ARID1A exon 20, G2069fs detected at higher MAF at progression (Supplementary Fig. 2). Additional clinical information including RECIST at restaging and time between sample collections are available in Supplemental Table 1.

Intrinsic ARID1A mutations in CRC have commonly been associated with microsatellite instability-high (MSI-H) and right-sided tumors[22]. Innocenti and colleagues previously reported that MSI-H patients have a shorter OS in the cetuximab arm compared to patients with microsatellite stable (MSS) or microsatellite instability-low (MSI-L) tumors[23]. Intriguingly, we observed 4/6 cetuximab-treated patients with acquired ARID1A mutations were initially diagnosed with left-sided tumors, with all 4 being MSS (Fig. 1c), suggesting MSS/MSI-H and sidedness are not a confounding factor for ARID1A mutations as a mechanism of acquired resistance to cetuximab treatment.

Taken together, these data indicate that KRAS and ARID1A mutations are enriched in patients treated with cetuximab but not bevacizumab, suggesting ARID1A mutations may also confer resistance to anti-EGFR therapy in 1 L mCRC patients treated with cetuximab.

### ARID1A mutation is a negative biomarker of anti-EGFR response

To further investigate the potential relevance of ARID1A mutations to the resistance to anti-EGFR therapies, we evaluated the association between pretreatment (intrinsic) ARID1A mutational status and outcome in cetuximab- vs. bevacizumab-treated patients. Available archival tissue from 562 participants of the CALGB/SWOG 80405 study were profiled with the FoundationOne® targeted assay[24]. We first corroborated that the biomarker evaluable population (BEP) is comparable to the intention to treat (ITT) population in terms of known baseline characteristics and outcome (Supplementary Table 2 and Supplementary Fig. 3a). Next, we examined overall survival (OS) and progression-free survival (PFS) for patients with pretreatment ARID1A alterations with known or likely functional impact (mutant) vs. ARID1A WT patients. Only the 429/562 patients with clinical data and at least one alteration detected in the tissue processed by Foundation Medicine (FMI) were considered for this analysis as reported previously[24]. Of these patients, 41/429 (9.5%) had known or likely pathogenic ARID1A alterations. This prevalence is consistent with the ~10% prevalence from previous reports of ARID1A mutation prevalence in mCRC[25,26]. ARID1A-mutant patients exhibited shorter OS when treated with cetuximab relative to bevacizumab (Log-Rank OS $p^{adj} = 0.001$, Fig. 2a; PFS $p^{adj} = 0.5$, Supplemental Fig. 3b). The interaction effect between ARID1A mutation status and treatment group was significant (OS, $p = 0.004$; PFS, $p = 0.02$) with the difference being observed in a multivariable model with adjustment for established clinical variables, PCR-based eRAS/BRAF mutation status, and chemo protocol (FOLFOX vs. FOLFIRI) (see methods) for OS (Supplemental Fig. 4, $n_{ARID1A\_mutant} = 41$, multivariable OS $p = 0.002$, hazard ratio (HR) 3.3, 95% confidence interval (CI) [1.6–6.8]) and for PFS (Supplemental Fig. 5, $p = 0.004$, HR 2.7 CI [1.4, 5.5]). In contrast, no difference was observed in OS and PFS for ARID1A-WT patients treated with cetuximab relative to bevacizumab [OS, multivariable $p = 0.74$, HR 1.0 (0.83–1.3), PFS, multivariable $p = 0.14$, HR 1.2 (0.95–1.5)].

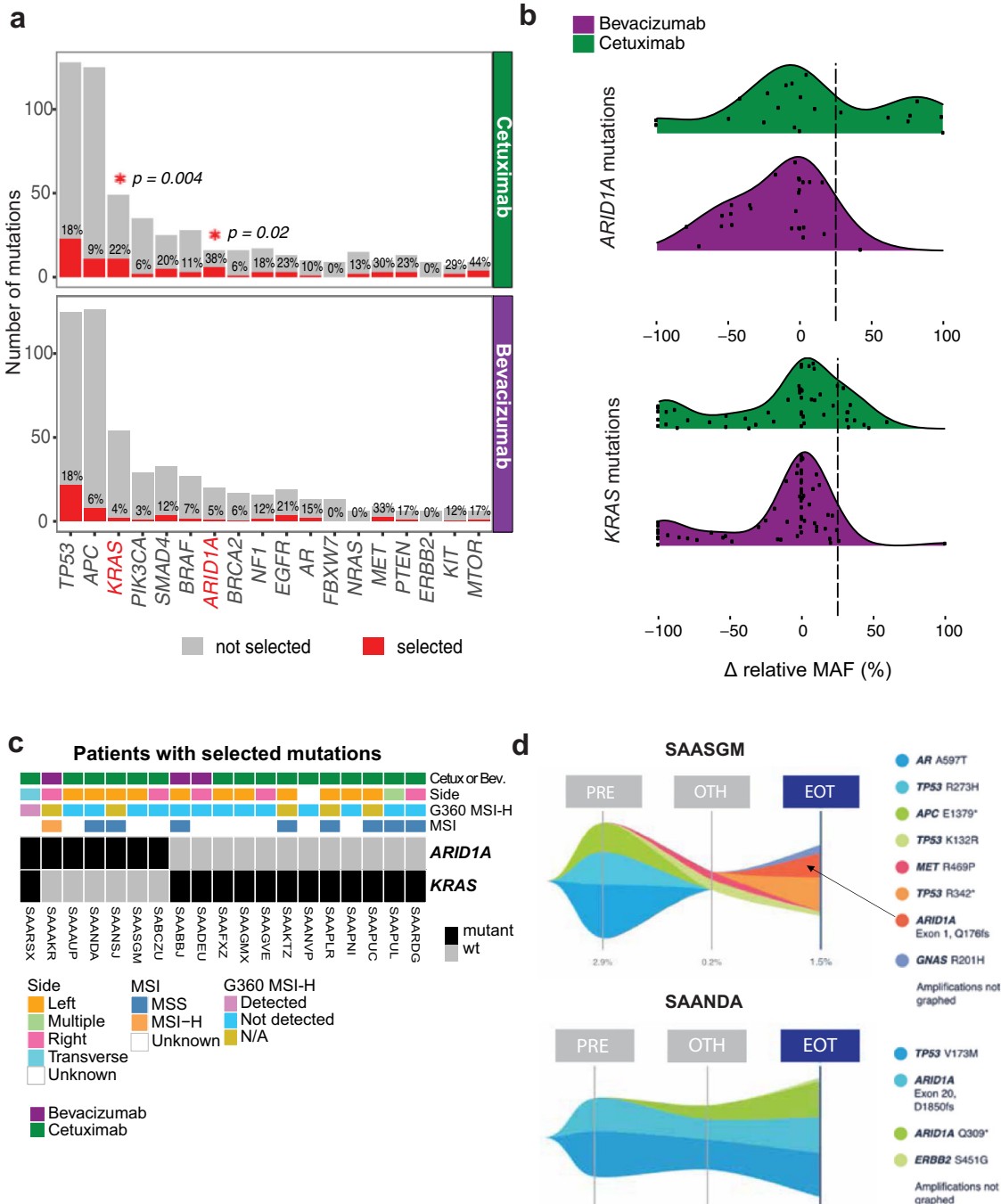

**Fig. 1 | ARID1A and KRAS mutations are enriched in patients treated with cetuximab. a** Number of patients with genomic alterations at either timepoint per treatment arm. Genomic alterations include non-synonymous single-nucleotide variants (SNV), indels, and gene rearrangements. Layered above the red bar is the corresponding percentage of patients with selected alterations, defined as an increase by 25 percentage points in the normalized MAF, calculated relative the highest reported MAF, between paired baseline and end of study cfDNA from 333 1L mCRC patients treated with either bevacizumab or cetuximab in combination with chemotherapy in the CALGB/SWOG 80405 trial. Asterisks indicate statistically significant differences (FDR *p*-value < 0.2) by two-tailed Fisher's exact test, restricted to 18 genes with at least 2 selected alterations in either arm. **b** Distribution of the % change in relative MAF between baseline and end of study for all KRAS and ARID1A alterations detected in all patients. Dashed line represents 25% threshold used to call tumors with selected mutations. **c** Clinical information for patients with selected ARID1A and/or KRAS alterations. **d** Representative mutation evolution maps for cetuximab-treated patients with selected ARID1A mutations. The maximum observed % MAF is shown at the bottom for each timepoint. PRE, at baseline; OTH, on treatment; EOT, at progression or end of study.

To assess the association between ARID1A intrinsic mutations and worse outcome in anti-EGFR vs. bevacizumab-treated patients in an independent "real-world" dataset, we investigated the relationship between ARID1A mutational status and outcome in anti-EGFR (cetuximab or panitumumab)- vs. bevacizumab-treated patients from the nationwide (US-based) de-identified Flatiron Health-Foundation

Medicine CRC clinico-genomic database (FH-FMI CGDB)[27–29]. The de-identified data originated from approximately 280 US cancer clinics (~800 sites of care). For consistency with CALBG80405, we defined anti-EGFR patients as those who received cetuximab or panitumumab but no bevacizumab at 1L (i.e., excluding combo treatment patients) vs. bevacizumab-treated patients as those who received bevacizumab

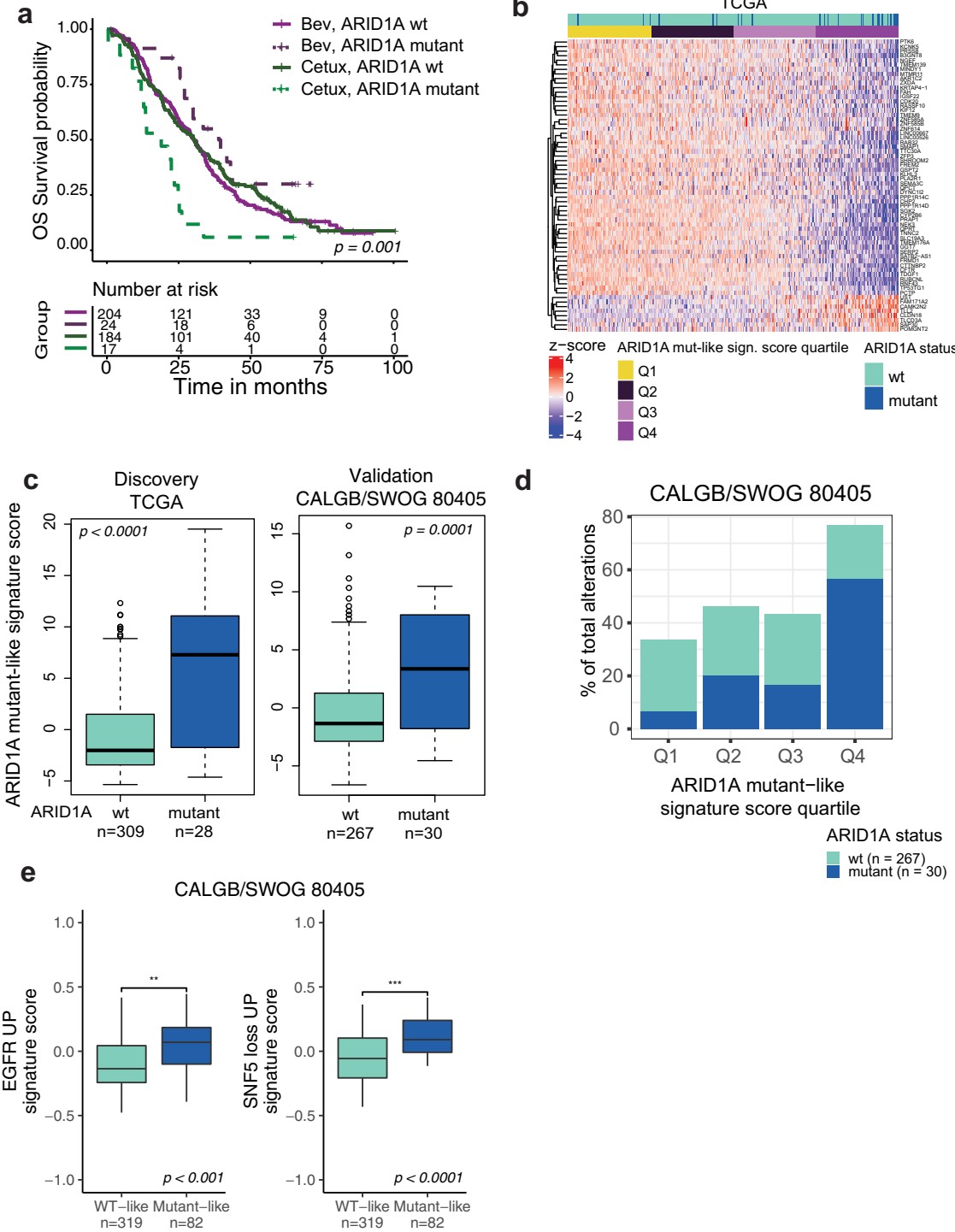

**Fig. 2 | ARID1A mutations confer poor outcome to cetuximab relative to bevacizumab possibly by partially inducing an EGFR transcriptional program.**
**a** Kaplan–Meier curves showing OS for 429 CALGB/SWOG 80405 patients with archival tissue available for targeted sequencing[24], stratified by ARID1A mutation status and treatment arm. Patients with ARID1A alterations classified as known or likely were considered mutant. *p*-values from log-rank test shown. **b** Heatmap representation of the ARID1A mutant-like signature in 578 CALGB/SWOG 80405 patients with gene expression data. **c** Distribution of the ARID1A signature score in the discovery TCGA cohort and validation cohort CALGB/SWOG 80405 cohort stratified by ARID1A mutation status. *p*-values determined by two-sided Wilcoxon

rank sum test. **d** ARID1A mutant-like signature score split into quartiles with the % of alterations in each quartile that were ARID1A mutant and WT in the 297 patients with both RNAseq and FMI mutation data available from the CALGB/SWOG 80405 cohort. **e** Distribution of MSigDB C6 Oncogenic signatures (GSVA scores) by ARID1A mutant-like signature group in CALGB/SWOG 80405. Two-sided Wilcoxon rank sum test; *p* < 0.05, ***, *p* < 0.001. **c, e** The interquartile range (IQR) is depicted by the box with the median represented by the center line. Whiskers maximally extend to 1.5× IQR (with outliers shown). Source data are provided as a Source Data file.

without any anti-EGFR at 1 L. By looking at the effect of ARID1A mutation status on the unadjusted OS estimates of eRAS/BRAF WT patients, we found ARID1A mutants have a median OS of 16 months when treated with cetuximab or panitumumab relative to 41 months when treated with bevacizumab (Supplemental Fig. 6a). Although the interaction effect between the treatment group and ARID1A mutation was not significant ($p = 0.5$), the main effect of ARID1A mutation status on anti-EGFR treatment group was significant. The forest plot confirms the worse outcome for patients without any of known KRAS, NRAS, or BRAF mutation but with ARID1A mutant patients compared to ARID1A WT on cetuximab or panitumumab when controlling for confounding factors, including age, gender, and MSI/MSS status [$p = 0.04$, HR 2.2 (1.0–4.8), Supplemental Fig. 6b].

## ARID1A mutant-like transcriptional signature enriches for mutations in SWI/SNF complex

In addition to mutations, ARID1A function can be inactivated or impaired by diverse alternative mechanisms such as inhibition of transcription by promoter methylation[30] or by mutations in other SWI/SNF complex components[21]. To investigate the potential role of ARID1A deficiency phenotype beyond mutations in resistance to cetuximab therapy, we developed an ARID1A mutant-like signature in an attempt to capture the transcriptional profile characteristic of patients with mutant/deficient ARID1A activity. This was derived from the list of differentially expressed genes between ARID1A mutant and WT tumors in TCGA colorectal cohort (see methods). Since ARID1A mutations are enriched in BRAF-mutant tumors[22], we accounted for eRAS/BRAF status in our test for differentially expressed genes by adding eRAS/BRAF status and the interaction between eRAS/BRAF and ARID1A mutation status as covariates in our model. This enabled us to focus on genes with changes in expression between ARID1A-mutant and WT tumors that were independent of eRAS/BRAF status, and to capture ARID1A-specific changes that may only occur in the eRAS/BRAF-WT group. This analysis revealed 8 genes that are significantly higher expressed and 56 genes that are significantly lower expressed in ARID1A-mutant tumors (FDR < 0.05; Supplemental Data 2).

An ARID1A mutant-like signature score was then calculated for each patient in both the TCGA dataset (discovery cohort) and the CALGB/SWOG 80405 study (validation cohort) as the reverse sign of the first principal component of the scaled expression value of these 64 genes, with expression shown in Fig. 2b for 337 patients in the TCGA cohort, ordered by the ARID1A (TCGA derived) mutant-like signature score. ARID1A-mutant tumors from the discovery TCGA cohort had significantly higher signature scores than ARID1A-WT tumors ($p < 0.0001$, Wilcoxon rank sum test; Fig. 2c, left) with ARID1A mutants being most enriched in the top quartile of the signature scores in TCGA (Supplemental Fig. 7a) and CALGB/SWOG 80405 (Fig. 2d). Further, we confirmed our signature's ability to detect ARID1A mutant subjects in the 297 patients from the CALGB/SWOG 80405 cohort with both RNAseq and FMI mutation calls available ($p < 0.001$; Fig. 2c, right). This difference was also observed in tumors when stratified by eRAS/BRAF status (CALGB/SWOG 80405 ANOVA $p < 0.0001$, $F$-value = 17.7, degrees of freedom (df) = 3, ARID1A mutant vs. wt in eRAS/BRAF WT $p^{adj} = 0.0007$ and eRAS/BRAF-mutant $p^{adj} = 0.0002$; TCGA ANOVA $p < 0.0001$, $F$-value = 37.3, df = 3, ARID1A mutant vs. wt in eRAS/BRAF WT $p^{adj} = 0.5$ and eRAS/BRAF-mutant $p^{adj} < 0.0001$, Supplementary Fig. 7b). However, the distribution of these scores was in general higher in eRAS/BRAF-mutant patients, suggesting partial shared signaling between ARID1A-mutant and eRAS/BRAF-mutant tumors.

We also evaluated the relationship between the ARID1A mutant-like signature and other CRC features including MSI status and CMS subtype. The main difference between the two cohorts was lower frequency of MSI tumors in CALGB/SWOG 80405 (23 of 286) compared to TCGA (64 of 337) cohorts (Supplemental Fig. 8a, b, left). In TCGA patients, we observed higher ARID1A mutant-like signature

scores in ARID1A mutant vs. WT tumors in both MSI and MSS subgroups (ANOVA $p < 0.0001$, $F$-value = 18.1, df = 3; wt_MSS-mut_MSS $p^{adj} < 0.0001$, wt_MSI-mut_MSI $p^{adj} = 0.1$; Supplemental Fig. 8a, b, right). In contrast, although CALGB/SWOG 80405 showed a similar trend of higher ARID1A mutant-like signature scores in ARID1A mutants vs. WT, these were not statistically significant when grouped by MSI status (ANOVA $p < 0.0001$, $F$-value = 41.2, df = 3; wt_MSS-mut_MSS $p^{adj} = 0.5$, wt_MSI-mut_MSI $p^{adj} = 0.8$) possibly due to the low number of MSI tumors in the CALGB/SWOG 80405 cohort.

In contrast, the prevalence of each CMS subtype profiled using the CMScaller (version 0.99.2)[31] was comparable between both cohorts (Supplemental Fig. 9a, b), with higher ARID1A mutant-like signature scores observed in CMS subgroups with higher prevalence of ARID1A mutations (CMS1, CMS3, and CMS4). There was also no impact of chemo protocol (FOLFOX vs FOLFIRI) on the ARID1A mutant-like signature despite 10/66 ARID1A mutant tumors were treated with FOLFOX vs 17/210 treated with FOLFIRI (Supplemental Fig. 10).

We subsequently assessed the ARID1A mutant-like signature in the context of functional alterations in other SWI/SNF complex members with selected functional events available from Mina et al.[32]. We detected higher signature scores in patients from both cohorts whose tumor was mutant for SWI/SNF complex members ($p < 0.001$, Wilcoxon rank sum test; Supplementary Fig. 11a). These included ARID2, PBRM1, SMARCA4, and ARID1B, in the TCGA cohort (Supplementary Fig. 11b). Altogether, SWI/SNF-mutant tumors had significantly higher signature scores than SWI/SNF-WT tumors. Therefore, these data indicate that the ARID1A mutant-like signature enriches for not only ARID1A mutations, but also for other mutations that affect the integrity and biological function of the SWI/SNF complex, suggesting that the transcriptional impact of mutations in the SWI/SNF complex beyond specific ARID1A mutations is effectively captured by our signature.

## ARID1A mutant-like signature enriches for a transcriptional program consistent with high EGFR-pathway activity and loss of SWI/SNF activity

To further evaluate the transcriptional changes affected by ARID1A deficiency, we calculated the ARID1A mutant-like signature for each sample and correlated these scores with the computed per-sample signature scores for all oncogenic signatures (C6) signatures from the Molecular Signature Database[33], in the TCGA ($n = 337$) cohorts. The curated C6 gene-set collection represents signatures of cellular pathways derived from perturbation experiments with genes often dysregulated in cancer. Two of the oncogenic signatures associated with the ARID1A mutant-like signature included genes upregulated by EGFR activity in MCF-7 breast cells stably overexpressing ligand-activatable EGFR[34] and genes upregulated upon the loss of SWI/SNF activity in SNF5 knockout murine embryonic fibroblast (MEF) cells[35], both correlated with the ARID1A mutant-like signature regardless of eRAS/BRAF mutation status in TCGA and CALGB/SWOG 80405 cohorts ($p^{adj} < 0.0001$, Pearson correlation > 0.4; Supplementary Fig. 12a, b). Beyond these gene expression signature correlates, ARID1A mutations were enriched in patients with ARID1A mutant-like signature group (Q4, top quartile enriched in ARID1A mutants) vs WT-like signature group (Q1–Q3) in both the TCGA (Supplementary Fig. 12c) and CALGB/SWOG 80405 cohort (Fig. 2e). Taken together, these results suggest the ARID1A mutant-like signature captures not only transcriptional gene expression changes consistent with the expected loss of SWI/SNF function but also with active EGFR signaling and downstream cell proliferation.

## ARID1A mutant-like transcriptional signature predicts reduced efficacy in cetuximab-treated patients and patient-derived xenograft models

After having identified a signature that captures defects in the SWI/SNF complex, we sought to assess the predictive value of this

signature for response to cetuximab compared to bevacizumab. Since eRAS/BRAF-mutant patients are expected to respond poorly to cetuximab[6,36], we evaluated clinical outcome by signature group defined by ARID1A mutant-like signature scores stratified by quartile (Q4, ARID1A mutant-like; Q1–Q3, WT-like) in the eRAS/BRAF WT patients from the CALGB/SWOG 80405 study. Similar to ARID1A-mutant tumors in Fig. 2a, the 25 ARID1A mutant-like eRAS/BRAF WT patients had shorter OS when treated with cetuximab compared to bevacizumab (OS, Fig. 3a; PFS, Supplemental Fig. 13). The interaction effect between ARID1A mutation signature and treatment group was significant (OS, $p = 0.007$; PFS $p = 0.001$) with the difference being observed in a multivariable model adjusting for clinical covariates and chemo protocol [$n_{ARID1A\_mutant-like} = 25$, OS, Supplemental Fig. 14, multivariable $p = 0.001$, HR 5.0 (1.9–13); PFS, Supplemental Fig. 15, $p = 0.007$, HR 3.4 (1.4–8.3)]. Patients with ARID1A mutant-like tumors were also less likely to respond (complete or partial confirmed best response) than patients with WT-like tumors when treated with cetuximab, and the proportion of overall confirmed best response rate (ORR) was significantly lower in the ARID1A mutant-like vs. WT-like group (Fisher exact $p = 0.006$, ORR = 44% vs. 89%, respectively; Fig. 3b). There was no significant difference in the ORR between the ARID1A mutant-like and WT-like groups for bevacizumab (Fisher exact $p = 0.5$, ORR = 50% vs. 63%, respectively). These data are consistent with the concept that the ARID1A mutant-like signature predicts resistance to cetuximab but not bevacizumab.

We further evaluated the ability of the ARID1A-mutant-like signature to predict decreased sensitivity to anti-EGFR treatment in an independent cohort of 244 cetuximab-treated patient-derived xenografts (PDX) with available response and gene expression data[37]. Since ARID1A genetic or mutation status was not available, we assigned an ARID1A mutant-like signature score to each PDX tumor, and stratified the 132 eRAS/BRAF WT tumors into an ARID1A mutant-like vs. WT-like group based on the top quartile as previously described in Supplementary Fig. 7a. Consistent with our previous observations, for eRAS/BRAF WT PDX tumors, ARID1A mutant-like tumors were less responsive to cetuximab than the ARID1A WT-like group as evidenced by a greater tumor volume increase observed at 3 weeks post-treatment (Wilcoxon rank sum $p < 0.01$, Fig. 3c), as well as by a lower fraction of responder cases (PR, SD-PR) in the ARID1A mutant-like group compared to the ARID1A WT-like group (Fisher exact $p = 0.05$, ORR = 11% vs. 36%, respectively, Fig. 3d). As a positive control, similar observations were made in cetuximab-resistant KRAS mutant tumors, as seen by a greater tumor volume increase observed at 3 weeks post-treatment ($p < 0.001$, Fig. 3e), as well as by a lower fraction of responder cases (PR, SD-PR) in the KRAS mutant-like group compared to the KRAS WT-like group (Fisher exact $p < 0.0001$, ORR = 2% vs. 28%, respectively, Fig. 3f).

### ARID1A mutations and EGFR/MAPK-pathway alterations are mutually exclusive in lung cancer and CRC patients

To gain further pathway and functional insight into possible genetic interactions underlying ARID1A deficiency-driven resistance to anti-EGFR therapy, we next searched for oncogenic dependencies or mutual exclusivities between ARID1A mutations and other alterations in cancer. We utilized the FoundationCORE® database[38] with targeted DNA sequencing data on 16,931 CRC tumors to identify other mutations that are either mutually exclusive or co-occurring with ARID1A mutations. We applied the SELECT algorithm[32] to all reported mutations and copy-number alterations with known or likely oncogenic significance, and adjusted for MSI status to remove its effect on the prevalence of detected alterations. Since EGFR is infrequently mutated in treatment-naïve CRC (0.5%)[25], we also performed this analysis in the 29,757 FoundationCORE® lung cancer samples, a tumor type where EGFR and ARID1A mutations are more frequently detected at baseline[39].

Patterns of co-occurrence in large patient cohorts often reflect functional synergies or evolutionary dependencies, whereas mutually exclusive events are likely functionally redundant or, less frequently, examples of synthetic lethality[32,40,41]. The SELECT algorithm revealed 23 genes in CRC and/or lung with alterations that were significantly mutually exclusive or co-occurring with ARID1A mutation (Fig. 4a). Strikingly, EGFR short variant (SV) was the top mutually exclusive alteration with ARID1A mutations in lung cancers (Fig. 4b). Of 5980 lung cancer patients with ARID1A and/or EGFR SVs, only 100 patients (1.7%) had both SVs (Fig. 4c). Most of the EGFR SVs in the FoundationCORE® lung data were activating mutations known to confer increased proliferation through sustained EGFR signaling (Fig. 4d, top). Furthermore, we also observed significant mutually exclusivity between EGFR amplification and ARID1A mutations in lung (FDR < 0.1; Fig. 4b, top) and a tendency for mutually exclusivity between ARID1A mutations and EGFR amplification (0.92%) in CRC (FDR < 0.1; Fig. 4b, bottom). No association with EGFR mutations was observed in CRC, likely due to low detection power given its low prevalence in this cohort (0.5%). Other co-occurring and mutually exclusive signaling pathway genes in CRC included BRAF, KRAS, SMAD4, AMER1, MEN1, and TERT (Fig. 4a). Other altered genes in Fig. 4a were enriched for WNT signaling in CRC, and for receptor tyrosine kinases that can cause overactivation of MAPK or PI3K pathways in lung cancer (hypergeometric test $q < 0.05$; Supplementary Data 3 and 4). Therefore, these data suggest that ARID1A mutations may be functionally redundant with MAPK and EGFR signaling in lung cancers and CRC.

## Discussion

In CRC, baseline pre-existing KRAS mutations provide intrinsic resistance to anti-EGFR containing therapies[6,36]. During treatment, the selective pressure induced by EGFR blockade also promotes acquired resistance, through expansion of tumor clones with mutations in KRAS[18,19], EGFR (extracellular domain, ECD)[3], and less frequently alterations in NRAS, BRAF, MEK1, and NF1[8,42] in addition to other lower prevalence alterations.

In this study, by analyzing cfDNA at baseline and end of treatment from the CALGB/SWOG 80405 study, we found that KRAS and ARID1A mutations were positively selected in a subset of the patients treated with cetuximab- but not bevacizumab-containing regimens. These observations may be explained by: (i) evaluation of a broader set of actionable mutations including ARID1A beyond canonical alterations downstream of EGFR such as KRAS/BRAF/NRAS/PI3K, (ii) the large number of patient samples analyzed to detect a low prevalent acquired mechanism of resistance such as ARID1A, and (iii) the ability to compare with a control arm (CALGB/SWOG 80405 was a randomized study). The reasons for not detecting other known prevalent emerging mutations such as EGFR (ECD) in this study are unclear, but could be related to the relative limited genomic heterogeneity and tumor burden in first-line (this study) compared to later lines of treatment as EGFR ECD mutations are thought to take typically longer than KRAS mutations to emerge[17] and are not likely to be founding clonal events[3,4]. We also did not identify mutations preferentially enriched in the bevacizumab-treated patients. The limited number of genes covered on the mutational panel coupled with the bias toward actionable pan-cancer targets may also have limited our ability to discover novel bevacizumab resistance mutations.

Intrinsic ARID1A mutations are typically seen in right-sided CRC tumors and associated with MSI-H status and BRAF mutations. In contrast, the acquired ARID1A mutations detected in this study were mainly from patients with left-sided (4/6), MSS (5/6) primary tumors, and were not associated with BRAF alterations (6/6). This "unexpected" high proportion of left-sided ARID1A secondary mutations is consistent with the known increased efficacy (and stronger selective pressure) of EGFR blockade in left-sided tumors (CALGB data, NCI guidelines). Finally, we did not observe co-occurrence of ARID1A cetuximab-selected mutations with KRAS or other alterations known

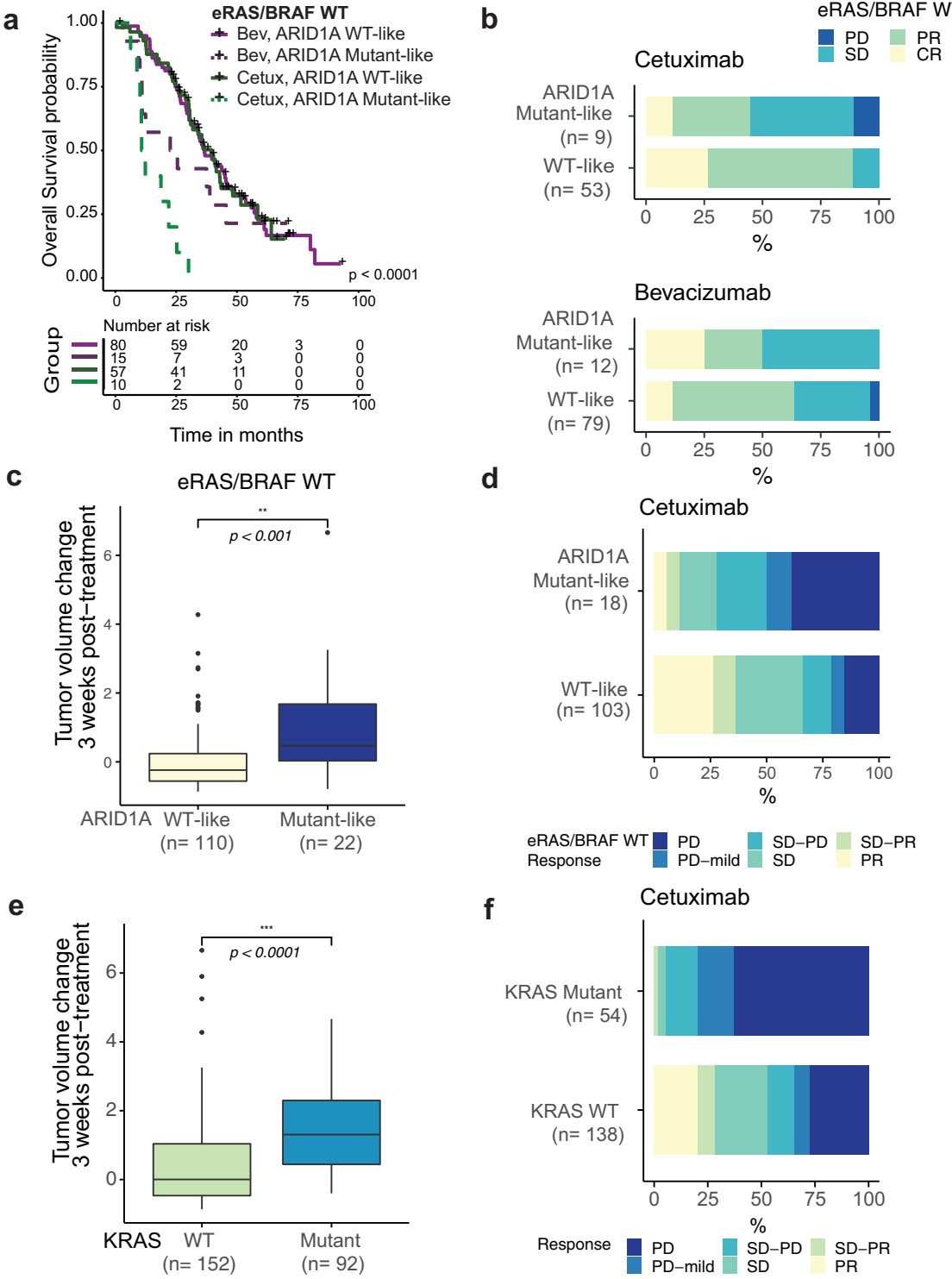

**Fig. 3 | ARID1A mutant-like signature enriches for cetuximab-resistant patients in eRAS/BRAF WT tumors. a** Kaplan–Meier curves showing OS in $n = 162$ eRAS/BRAF WT patients stratified by ARID1A signature group (mutant-like vs. WT-like), using the signature score Q4 cutoff, and by treatment arm with $p$-values from log-rank test shown. **b** Clinical response (best RECIST) to cetuximab (*top*) and bevacizumab (*bottom*) was stratified by ARID1A mutant-like signature group for the $n = 153$ CALGB/SWOG 80405 eRAS/BRAF WT patients with RECIST response, RNAseq and FMI alterations data available. PD, progressive disease; SD, stable disease; PR, partial response; CR, complete response. CR and PR cases were considered responders and SD and PD cases non-responders to treatment. **a**, **b** Source data are provided as a Source Data file. **c**, **e** Change in tumor volume from baseline to after 3 weeks on cetuximab treatment (20 mg/kg, twice a week) for 132 eRAS/BRAF WT PDX models shown by ARID1A signature group (**c**), and for all 244 PDX models shown by KRAS mutation status (**e**). $p$-values by two-sided Wilcoxon rank sum test (***$p < 0.001$). The interquartile range (IQR) is depicted by the box with the median represented by the center line. Whiskers maximally extend to 1.5× IQR (with outliers shown). **d**, **f** Cetuximab response by ARID1A signature group in the eRAS/BRAF WT population (**d**; $n = 121$) and by KRAS status in the full PDX cohort (**f**; $n = 192$).

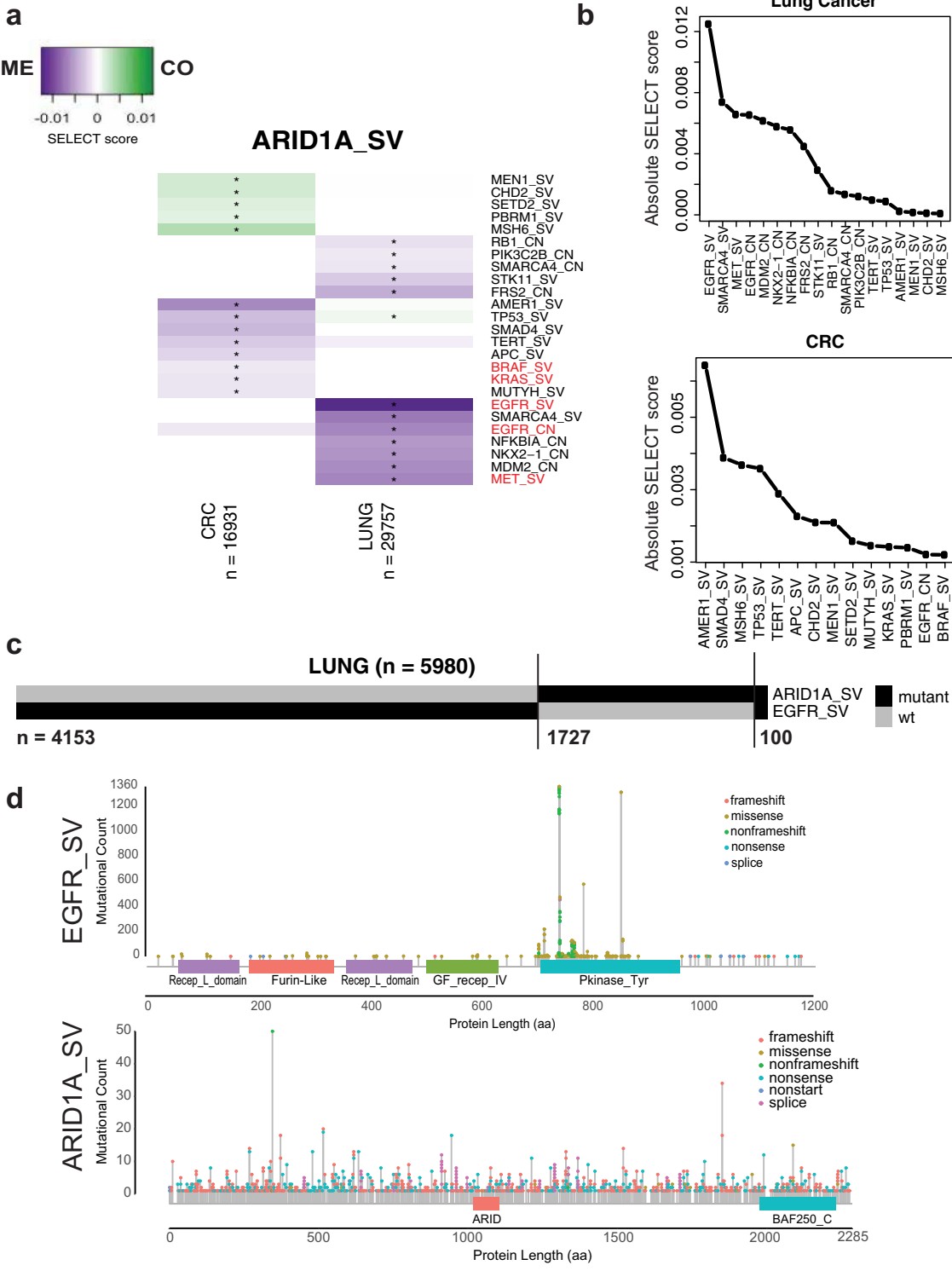

**Fig. 4 | ARID1A mutations may compensate for mutant EGFR signaling by partially inducing an EGFR transcriptional program. a** Overview of genes with SV or copy-number alterations (CN) that are mutual exclusive (ME, purple) or co-occurring (CO, green) with ARID1A SVs based on the SELECT method[32], in a CRC (n = 16,931) and lung cancer (n = 29,757) cohort from the FMI FoundationCORE database[38]. Known and likely oncogenic alterations in each gene were considered for the SELECT analysis. Asterisks indicate significant relationships (FDR < 0.1). **b** Overview of significant interactions with ARID1A SVs detected in lung cancer (*top*) and CRC (*bottom*), ranked by SELECT scores. **c** Mutation occurrences for 5980 patients with ARID1A and/or EGFR mutations in the FMI lung cancer cohort. **d** Overview of known and likely EGFR (*top*) and ARID1A (*bottom*) mutations in lung cancer patients. aa amino acid.

to cause resistance to cetuximab treatment. Hence, progression on cetuximab in patients with a selected ARID1A mutation is unlikely to be attributed to co-mutation in other known resistance mechanisms, and thus ARID1A selected mutations may independently contribute to resistance to anti-EGFR therapy.

ARID1A is the most frequently mutated subunit of the SWI/SNF chromatin remodeling complex in cancer[43] and is frequently altered in CRC, non-small cell lung cancer (NSCLC), clear cell ovarian cancer, hepatocellular carcinoma, and endometrial cancer among other tumor types[44–49]. ARID1A mutations have been previously implicated in tumor

progression, chemoresistance, metastatic spread and activation of oncogenic signaling pathways such as PI3K and MAPK. The vast majority of mutations in ARID1A are truncating mutations[50], which are thought to promote cancer by destabilizing the SWI/SNF complex to facilitate cell growth[51]. Since the SWI/SNF complex modulates DNA accessibility for cellular processes involved in chromatin structure including transcription, DNA replication and repair, loss of ARID1A is thought to globally deregulate and impact gene transcription[52,53]. The specific molecular mechanisms by which ARID1A mutations contribute to cancer are likely to be complex and remain an area of active investigation. In this context, the emergence of ARID1A mutations in cetuximab-treated patients indicates a previously unsuspected functional link between EGFR signaling and ARID1A/SWI/SNF effectors in CRC patients. Although the biological convergence between these two pathways is not necessarily evident a priori, a recent functional genetic screen in a NSCLC cell line identified loss of function ARID1A and other SWI/SNF components as important mechanisms of resistance to EGFR inhibition in vitro[54]. Our paper shows that this mechanism of resistance mediated by ARID1A/SWI/SNF loss that was initially uncovered by functional genetic screens in vitro is likely to be relevant in human patients treated with agents that inhibit EGFR signaling.

If secondary ARID1A mutations are indeed a mechanism of acquired resistance to cetuximab then intrinsic mutations in this gene may also predict primary resistance to anti-EGFR treatment. By performing correlative analysis between ARID1A status at diagnosis in tissue and outcome in CALGB/SWOG 80405 we were able to show that patients with known or likely intrinsic mutations in ARID1A have indeed worse overall survival on cetuximab (independently of MSI or eRAS/BRAF status) than bevacizumab while this differential outcome was not observed in ARID1A WT patients. We confirmed these observations using real-world data. One possible limitation for this interpretation is that, given its low prevalence in CRC, the number of ARID1A mutants included in these correlative analyses is relatively small. Together these findings suggest that both primary and secondary ARID1A mutations can compensate for the lack of EGFR signaling. Although the prevalence of ARID1A in CRC is relatively low (~5–11%)[25,55], this finding, if validated in a prospective study, may have important clinical implications. Our data suggest that ARID1A mutant patients could be potentially excluded from cetuximab treatment. However, ARID1A mutation by itself may be an imperfect biomarker, as ARID1A downstream function can be impaired by other mechanisms besides mutation.

In an attempt to more broadly characterize these downstream biological consequences at the transcriptional level, we leveraged genomic and gene expression data to discover and validate an ARID1A loss of function-like gene expression signature that not only identifies ARID1A mutant tumors but also more broadly other tumors with likely impairment of the SWI/SNF complex. By applying this signature, we found that ARID1A mutant-like tumors were less responsive to anti-EGFR treatment than the ARID1A WT-like group in CALGB/SWOG 80405. This was also confirmed in eRAS/BRAF WT PDX tumors. Thus, the downstream biological consequences of ARID1A loss of function can also be defined at the phenotypic/transcriptional level. Our data suggest that CRC tumors with functional impairment of ARID1A function (as defined by the ARID1A signature) may also be resistant to anti-EGFR therapies, thus indicating that the prevalence of this phenotype may be broader than what ARID1A genomic mutation data would suggest.

Our data showing that the ARID1A mutant-like signature enriches in transcriptional programs consistent with active EGFR and loss of SWI/SNF activity, suggests that the ARID1A/SWI/SNF complex negatively modulates transcription of EGFR downstream effectors. EGFR signaling is known to promote sensitivity to anti-EGFR therapies[8,56–58]. However, the role of the EGFR pathway in the context of loss of ARID1A/SWI/SNF activity has not been characterized in these studies.

The transcriptional convergence between EGFR activation and ARID1A loss provides a likely mechanistic explanation for the contribution of ARID1A mutations to the resistance to anti-EGFR therapy. A detailed molecular characterization of the interaction between these two pathways is beyond the scope of the present manuscript but warrants further investigation.

In CRC patient tumor genomic data, we were able to detect a trend toward a mutually exclusive interaction between ARID1A and rare baseline EGFR amplifications, as well as significant mutually exclusive interactions between ARID1A and BRAF and KRAS, two genes that are downstream of EGFR signaling and known to confer resistance to anti-EGFR therapy. Data from NSCLC patients, in which both EGFR and ARID1A are frequently mutated, confirmed the relevance of this mutually exclusive genetic interaction between these two genes in the context of EGFR dependent tumors. Thus, the mutual exclusive interaction described here provides additional independent confirmation of the relevance of the functional link between ARID1A and EGFR/MAPK-pathway during tumor progression in CRC and NSCLC patients that are EGFR signaling dependent that is consistent with previously published genetic data in NSCLC cell lines.

The data presented in this manuscript add to the large and growing body of clinical evidence indicating that most CRCs are addicted, in one way or another, to EGFR/MAPK activation. This is evidenced by the efficacy of cetuximab in WT patients, by the secondary mutations in this pathway emerging in response to anti-EGFR blockade, by the high prevalence of KRAS mutations, and by the sensitivity of BRAF V600E mutations to combinations of cetuximab and BRAFV600E inhibitors. Our results not only indicate that ARID1A could be potentially used, if further confirmed, as a predictive biomarker for excluding patients from cetuximab treatment but also provide a rationale for exploring therapeutic MAPK inhibition in a previously unsuspected genetically defined segment of CRC patients.

## Methods

The presented research complies with all ethical regulations. Institutional review board approval was required at all participating centers, which are listed at https://clinicaltrials.gov/ct2/show/NCT00265850, and all participating patients provided written informed consent. Patients were enrolled at centers across the National Cancer Trials Network in the United States and Canada. CALGB is now part of the Alliance for Clinical Trials in Oncology. The study was conducted in accordance with recognized ethical guidelines of the Declaration of Helsinki, CIOMS, Belmont Report, and the U.S. Common Rule. The clinical trial number for this trial is NCT00265850.

### CALGB/SWOG 80405 study design, patient cohort, and response assessment

Samples for this analysis were collected from the CALGB/SWOG 80405 trial, which was a phase III trial designed to determine whether the addition of cetuximab compared to bevacizumab to chemotherapies was superior as first-line treatment in advanced or metastatic CRC. The study design, patient eligibility, and clinical outcome of this trial have been reported previously[23,59]. Additional information about the study is available at ClinicalTrials.gov identifier: NCT00265850. Best clinical response was determined according to RECIST 1.1 (Sum of the longest diameters of the target lesions on CT scans). The number of patients enrolled in the trial and the blood samples used for analysis are reported in Supplementary Fig. 16. Informed consent was obtained from all patients for analysis conducted in this study.

### Nucleic acid sample preparation

The pathologic diagnosis of each case was confirmed by review of hematoxylin and eosin (H&E) stained slides. Tumors marked by a pathologist based on H&E images were macro-dissected for DNA and

RNA extraction. Tumor DNA was extracted by QIAamp DNA formalin-fixed, paraffin-embedded tissue kits (QIAGEN, Hilden, Germany). Total RNA was isolated using Roche High Pure FFPE RNA Micro Kit (Roche Applied Science, Mannheim, Germany).

## Mutation profiling (FoundationOne® and allele-specific PCR)

DNA extracted from 562 patients was submitted to Foundation Medicine (Cambridge, MA) for targeted next-generation sequencing (NGS)-based genomic profiling following standard procedure as previously described[60,61]. Adaptor-ligated DNA underwent hybrid capture for coding exons of 395 cancer-related genes plus select introns from 31 genes frequently arranged in cancer FoundationOne® panel as previously reported[24]. In addition, mutation hotspots in AKT, APC, BRAF, CTNNB1, EGFR, FBXW7, HRAS, KRAS, NRAS, PIK3CA, and TP53 were detected by allele-specific PCR from 843 patients, and the detected mutations in those genes have been previously reported[23]. Unless otherwise indicated, FoundationOne®-based mutations were used. For some analyses, KRAS, NRAS and BRAF hotspot mutations from additional samples were included, as indicated. In addition, comprehensive genomic profiling (CGP) was carried out for 16,931 CRC and 29,757 lung tumors in a Clinical Laboratory Improvement Amendments (CLIA)-certified, CAP (College of American Pathologists)-accredited laboratory (Foundation Medicine Inc., Cambridge, MA, USA) on all-comers during the course of routine clinical care. Approval was obtained from the Western Institutional Review Board (Protocol No. 20152817). Hybrid capture was carried out for coding exons from at least 324 cancer-related genes plus select introns from up to 31 genes frequently rearranged in cancer. We assessed all classes of genomic alterations (GA) including short variant (SV), copy-number (CN), and rearrangement alterations[62].

## Gene expression profiling

Whole-transcriptome profiles were generated for 578 patients using TruSeq RNA Access technology (Illumina®). RNAseq reads were first aligned to ribosomal RNA sequences to remove ribosomal reads. The remaining reads were aligned to the human reference genome (NCBI Build 38) using GSNAP version "2013-10-10", allowing maximum of two mismatches per 75 base sequence (parameters: '-M 2 -n 10 -B 2 -i 1 -N 1 -w 200000 -E 1—pairmax-rna=200000—clip-overlap). To quantify gene expression levels, the number of reads mapped to the exons of each RefSeq gene was calculated in a strand-specific manner using the functionality provided by the R package GenomicAlignments13 (Bioconductor). RNAseq analysis was performed in R using limma (version 3.53.5) Bioconductor package. Heatmaps were generated with ComplexHeatmap (version 2.13.0) Bioconductor package.

## cfDNA analysis

For all patients who enrolled in the CALGB80405 study and consented to have their blood be kept for future unknown use in research to learn about, prevent, treat or cure cancer, one 5 mL citrate tube and three 5 mL EDTA tube were drawn prior to the initiation of protocol therapy (baseline), at first restaging (8 weeks, during treatment), and at the discontinuation of protocol therapy. After centrifuging for 10 to 15 min at 1300 x $g$ (or in accordance with collection tube manufacturer's instruction), plasma samples were aliquoted into 1.8 mL cryovials at 0.5 mL per vial, then were processed and frozen within 3 h of collection and stored at −80 °C. For cfDNA analysis, a total of 953 EDTA or citrated plasma specimens (1–2 mL) from the CALGB/SWOG 80405 study collected at different time points (baseline: 354; during treatment: 254; at the end of protocol therapy: 345) were analyzed using the Guardant 360 cfDNA assay. The Guardant 360 assay is a Clinical Laboratory Improvement Amendments (CLIA)-certified targeted digital sequencing panel designed to detect SNVs, as well as selected insertions/deletions, amplifications, and fusions. cfDNA isolation and sequencing were performed by Guardant Health using previously described methodology[63–65]. Briefly, blood was processed upon receipt to isolate plasma by centrifugation at 1600 × $g$ for 10 min at 4 °C. Plasma was immediately aliquoted and stored at −70 °C. Cell-free DNA was extracted from 1 mL aliquots of plasma using the QIAamp circulating nucleic acid kit (Qiagen), concentrated using Agencourt Ampure XP beads (Beckman Coulter), and quantified by Qubit fluorometer (Life Technologies, Carlsbad, CA, USA). All cfDNA sequencing and analysis was performed at Guardant Health (Redwood City, CA, USA). Illumina sequencing reads were mapped to the hg19/GRCh37 human reference sequence, and genomic alterations in cfDNA were identified from Illumina sequencing data by Guardant Health's proprietary bioinformatics algorithms. These algorithms quantify the absolute number of unique DNA fragments at a given nucleotide position, thereby enabling circulating tumor DNA to be quantitatively measured as a fraction of total cfDNA.

## Real-world data analysis

The data presented in Supplementary Fig. 6 was derived from Flatiron (FH) and FMI, this is an observational, non-interventional cohort study. Retrospective longitudinal clinical data were derived from electronic health record (EHR) data, comprising patient-level structured and unstructured data, curated via technology-enabled abstraction, and were linked to genomic data derived from FMI comprehensive genomic profiling (CGP) tests in the FH-FMI CGDB by de-identified, deterministic matching[28]. Genomic alterations were identified via comprehensive genomic profiling (CGP) of >300 cancer-related genes on FMI's next-generation sequencing (NGS) test, i.e., FoundationOne® CDx, FoundationOne® or FoundationOne Liquid®[62]. Our study cohort includes 5511 CRC patients in the disease-specific FH-FMI CGDB who: (a) are in the Q3 2020 release; (b) have a metastatic CRC diagnosis (may be de novo metastatic or have progressed to metastatic disease from early stage CRC diagnosis). The primary endpoint for the Kaplan–Meier Analysis was OS, which is defined as the length of time (in months) from the start of the first-line of therapy (as the index date) until death from any cause or censoring date (we use the end of treatment date). Our analysis was restricted to patients with FMI CGP testing before start of treatment (1 L) for both alteration and WT cohorts. We compared anti-EGFR (cetuximab or panitumumab) treated patients to anti-VEGF therapy (bevacizumab) for patients with ARID1A mutation status available. ARID1A patients with short variant alterations annotated as known and likely were defined as ARID1A mutant positive. For the analysis, Cox proportional hazards (PH) regression models was applied for multivariable analyses to obtain the hazard ratio (HR) and 95% confidence intervals (CIs) while controlling for confounding factors, including age, gender, and MSI status, within the KRAS/NRAS/BRAF (known/likely) WT cohort. Based on the Schoenfeld residuals test results of each comparison group, the PH assumption holds and we do not need to stratify any covariates. KM curves show unadjusted OS with log-rank test used to assess differences between the groups. Multivariate Cox PH models were used to evaluate cetuximab or panitumumab compared to bevacizumab and the interaction with ARID1A status (mutant vs. wildtype) adjusted for confounding factors of age, gender, MSI status, and KRAS/NRAS/BRAF mutation status. We present the forest plot that shows HR (exp(coef) or the instantaneous relative risk) for patients receiving Cetuximab or Panitumumab from the Cox PH models.

## ARID1A mutant-like signature generation

The ARID1A mutant-like transcriptional signature was derived from the list of differentially expressed genes between ARID1A mutant and WT tumors in TCGA colorectal tumors. To enrich for mutations that would impact the function of the protein, TCGA mutant tumors were defined as those with alterations annotated as selected functional events (SFE) by Mina et al.[32]. These alterations were curated from the thousands of

observed copy-number alterations (CNAs) and somatic mutations observed in the TCGA cohort to enrich for potential driver mutations[32,41]. Transcriptional profiles of the 28 ARID1A mutant tumors (with SFE) were compared to the 309 ARID1A-WT tumors (i.e., without SFE) using the limma voom function in R, with eRAS/BRAF status and the interaction between eRAS/BRAF and ARID1A mutation status included as covariates, and genes with and FDR > 0.05 were selected as the mutant-like signature.

### ARID1A mutant-like signature validation

The ARID1A mutant-like signature comprises the differentially expressed genes between the ARID1A mutant and WT samples with MSI status, eRAS/BRAF status and the interaction between eRAS/BRAF and ARID1A status added as covariates (FDR < 0.05) with the voom function implemented in the limma R package. ARID1A mutation status, MSI and eRAS/BRAF status were all derived from the FMI targeted panel. ARID1A mutation status for TCGA CRC patients was obtained from the Pancan23 genomic alteration matrix of selected functional events available at http://ciriellolab.org/select/select.html[32]. The ARID1A mutant-like signature score was calculated as the reverse sign of the first principal component of the $z$-score transformed expression of each gene in the signature. For the PDX cohort, we calculated the ARID1A mutant-like signature score from the gene expression data available at GSE76402[58]. GEO data was downloaded using the GEOquery (version 2.65.2) Bioconductor package. In both cohorts, tumors were stratified into ARID1A mutant-like and WT-like groups based on quartile (Q4, mutant-like; Q1–Q3, WT-like). eRAS/BRAF status in the PDX cohort was based on the mutation calls reported for KRAS, BRAF and NRAS[58].

### SELECT and Gene-set enrichment analysis

The SELECT algorithm was used to look for co-occuring and mutually exclusive mutations in FoundationCore database. A copy of the algorithm can be downloaded from the Ciriello lab [http://ciriellolab.org/select/select_1.0.tar.gz]. The select algorithm (version 1.0)[32] was applied to the known and likely mutation calls from the FMI foundation core database for lung cancer and CRC patients. The hypergeometric test was used to compare the significance of the overlap between the genes identified by SELECT for CRC and lung, and Pathway Commons pathways. Enrichment $p$-values were corrected for the number of pathways tested using the Benjamini and Hochberg procedure. Complete results from the enrichment analyses are reported in Supplementary Data 3 and 4.

### Statistical analysis

The distribution of time-to-event end points was estimated by Kaplan–Meier curves. Associations with OS and PFS were tested using the log-rank test. Pairwise comparisons were corrected for multiple testing using the Benjamini & Hochberg method. A multivariable stratified Cox model was used to identify the association between a biomarker and time-to-event end points. Models were adjusted for age, treatment arm, sex, ECOG score, synchronous vs. metachronous metastases, number of metastatic sites, primary tumor location (right, transverse, left), MSI status, and eRAS/BRAF (KRAS, BRAF, and/or NRAS mutant vs. WT), while stratifying for prior adjuvant chemotherapy and prior radiation. For Cox models testing the effect of MSI status on ARID1A-mutant patients OS and PFS, models were adjusted for the clinical variables that were significant in univariable models: age, number of metastatic sites, side, and eRAS/BRAF mutation status (KRAS, NRAS, BRAF). All other statistical analysis and figures were generated in Rstudio IDE with R version 4.2.0.

### Reporting summary

Further information on research design is available in the Nature Research Reporting Summary linked to this article.

## Data availability

The TCGA publicly available RNAseq data used in this study are available in the Firebrowse portal (version 2016 01 28) http://firebrowse.org/. TCGA gene expression raw counts for colorectal cancer are available for download from https://gdac.broadinstitute.org/. The selected functional event mutation annotations for TCGA tumors are available from the Ciriello lab[32] http://ciriellolab.org/select/pancan23_dataset.zip. The PDX gene expression data used in this study are available in the GEO database under accession code GSE76402[58]. The CALGB/SWOG 80405 RNAseq data previously generated and used in this study are available in the GEO database under accession code GSE196576. The CALGB/SWOG 80405 ctDNA data generated in this study have been deposited in the dbGAP database under accession code phs002941. The data is available under restricted access due to them containing information that could compromise research participant privacy/consent. Access can be obtained by submitting a request to Xueping Qu (qu.xueping@gene.com). Individual-level data from the genotyped cohorts will be made available to researchers for academic purposes only following an approved analysis proposal for 1 year. Review timelines may vary but once approved access should be granted within a week. Real-world data in this study refers to observational data generated during routine clinical practice and collected outside regulated clinical trials by Flatiron Health and Foundation Medicine clinic-genomic database. Restrictions apply to the availability of the real-world data underlying the analysis for Foundation Medicine, Inc and Flatiron Health-Foundation Medicine CRC clinico-genomic database. For further details on Roche's Global Policy on the Sharing of Clinical Information, and how to request access to related clinical study documents, see https://www.roche.com/research_and_development/who_we_are_how_we_work/clinical_trials/our_commitment_to_data_sharing.htm. The remaining data are available within the Article, Supplementary Information or Source Data file. Source data are provided with this paper.

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

## Acknowledgements

The CALGB 80405 trial was supported by National Cancer Institute grants U10CA180821, U10CA180882, and U24CA196171 (Alliance for Clinical Trials in Oncology), U10CA180790 (Mayo Clinic), U10CA180791 (Memorial Sloan Kettering), 10CA180795 (Indiana University), U10CA180799 (University of Wisconsin), U10CA180830 (University of Southern California Norris Cancer Hospital), U10CA180836 (The University of Chicago), U10CA180838 (The University of North Carolina), U10CA180850 (The Ohio State University), U10CA180867 (Dana-Farber Cancer Institute), and U10CA180888 (SWOG). P30CA014089, UG1CA180830, UG1CA233253, UG1CA233327, UG1CA233373, U10CA180820 (ECOG-ACRIN). The trial was also supported in part by Bristol-Myers Squibb, Eli Lilly, Genentech, Pfizer, and Sanofi. We thank the CALGB80405 team and the patients participating in this trial. We thank all members of our group for valuable suggestions, critical review of the data and the manuscript, including Carin Espenschied at Guardant Health. The content is solely the responsibility of the authors and does not necessarily represent the official views of the National Institutes of Health. https://acknowledgments.alliancefound.org.

## Author contributions

F.I., H.-J. L., A.V., F.-S.O., O.K., X.Q., R.M.J, C.B. contributed to the overall study design. F.I., H.-J.L., A.V., F.-S.O., O.K., L.H., X.Q. collected and assembled all clinical biomarker data. R.M.J., F.-S.O. performed biomarker and statistical analyses. C.-F.L. and L.W. performed the real-world data analysis. R.M.J. established the ARID1A mutant-like signature score and conducted bioinformatic analyses. E.S. provided and analyzed data from Foundation Medicine database and generated Fig. 4d. N.I. generated mutation evolution maps in Fig. 1c, d and Suppl. Fig. 2a. R.M.J., C.B., A.D., X.Q., R.B., F.I., H.-J.L., A.V., O.Z., F.d.S.e.M, O.K., and F.-S.O. wrote and edited the manuscript. All authors contributed to data interpretation, discussion of results, and commented on the manuscript.

## Competing interests

A.P.V. reports consulting/advisory role from Taiho Pharmaceutical, Bayer AG, Halozyme, Eisai; research funding from Genentech, Roche, Bristol Myers Squibb; royalties from Now-UpToDate for authoring and maintaining two authors and travel expenses from Genentech, Roche, Halozyme, Bayer AG. F.I reports royalties from the Mayo Foundation on UGT1A1 testing. H.-J.L. is on the advisory Board/Consulting Genentech, Bristol Myers Squibb, Merck KG, Merck, Bayer, Oncocyte, Fulgent, G1 Therapeutics, Jazz Therapeutics, Biacara, Seattle Genetics, Abbvie, Amgen. E.S. Foundation Medicine, Roche; O. K. Zai Lab, San Francisco, CA; A. D. ORIC Pharmaceuticals; R. B., Freenome; A. V. Bristol Myers Squibb. Current affiliations of N.I.: Genentech, South San Francisco, CA; R.B.: Freenome, South San Francisco, CA; A.D.: ORIC Pharmaceuticals Inc., South San Francisco, CA; A.V.: Bristol Myers Squibb, Cambridge, MA; O. K., Zai Lab. and C.B.: Inhibrx, Inc, San Diego, CA. All other authors are employees and stockholders of Genentech/Roche.
