## [Peer Review File · Nature Communications]

ARID1A mutations confer intrinsic and acquired resistance to cetuximab treatment in colorectal cancerReviewers' Comments:

Reviewer #1:

Remarks to the Author:

The manuscript by Johnson et al. addressed longitudinal CRC cohorts treated with either cetuximab or bevacizumab, investigating mutations in cfDNA using the FM cancer panel. They show that a fraction of the patients that develop resistance to Cetuximab treatment display selected mutations in ARID1A, and that patients carrying intrinsic ARID1A mutations treated with cetixumab have a lower OS than the ones who received bevacizumab. The association of ARID1A mutations with treatment response in CRC is interesting and has important implications for the patient management. The authors report mutually exclusive genomic alterations in EGFR and ARID1A, however previous publications (e.g Karachaliou et al. 2018) mentioned that those could co-occur in lung carcinoma for instance. The second part of the manuscript regarding the EGFR-like expression signature is hard to follow and must be clarified. Besides, the methods pertaining to RNAseq are very short and lack important information. The accompanying Supplementary Table does not show directly gene expression values (minor remark, it displays 61 genes and not 64 as indicated in the text), nor the range of expression differences. It would be interesting to annotate the samples according to the different CRC tumor groups identified in previous studies, which might be relevant in the data interpretation. It is also unclear whether Folfox or FOLFIRI have a differential impact on the gene signatures. In the PDX section, the authors mention that the genetic information was not available, but it should be possible to extract it for the RNAseq data. Though there are indications of a potential connection between ARID1A mutations and the EGFR pathway, this is not clearly demonstrated in the present form of the manuscript. This study is interesting but could be strengthened.

Reviewer #2:

Remarks to the Author:

Johnson et al. perform targeted sequencing of cell-free DNA from colorectal cancer patients with drug resistant disease. They show that ARID1A mutations are enriched in emerging mutations from CRC patients treated with anti-EGFR (cetuximab) therapy. Although aspects of the research are not new, including the connections between ARID1A and EGFR/MAPK signaling, the work is a tour de force with high clinical and translational impact. In addition, the rigor of the study is strong and the results are conclusive. Therefore, I support publication.

Minor comments:

-ARID1A mutations are associated with tumor progression, chemoresistance and metastatic spread in cancer, and associations between ARID1A and MAPK in colorectal cancer have been reported. There should be more discussion on what's known in this regard. In addition, strong statements, such as, 'a connection between ARID1A mutations and EGFR/MAPK has not been previously reported,' should be omitted from the abstract, since there are published reports on this subject.

Reviewer #3:

Remarks to the Author:

The manuscript by Johnson et al identifies ARID1A as a mechanism of acquired resistance to antiEGFR based-therapy. Moreover, it describes ARID1A as a marker of primary resistance to antiEGFR therapy. The work is original and novel. However, although the manuscript provides evidence based on analysis of data from a randomized clinical trial and validation cohorts (including PDX and real-world data), the unfrequent percentage of the event (ARID1A mutation) limits the clinical interpretation and results should be cautiously interpreted. Moreover, because cetuximab/panitumumab/bevacizumab are administered together with chemotherapy, the impact of the chemotherapy scheme (irinotecan vs

oxaliplatin) may be a confounding factor. To add robustness to the causality of ARID1A in driving acquired and primary resistance to anti-EGFR therapy, functional studies in preclinical models (i.e., ARID1A overexpression confers resistance?, ARID1A knockdown reverts resistance?) are recommended.

Specific comments to improve the interpretation of data and conclusions are:

ctDNA is a very appropriate tool to monitor for the emergence of clones of resistance, however interpretation of the results, relative MAFs and MAF's cutoffs are not currently standardized. The authors should clarify how the 25% increase in the relative MAF is calculated (i.e., relative to normal cfDNA in the sample? Relative to total ctDNA in the sample? relative to the highest MAF?). That may completely change the results. For example in Figure 1d, in the second evolution map it is not clear that ARID1A increase is not a consequence of an increase in total ctDNA (minor comment: please use same color for each specific mutation in all figures). Moreover, authors should justify the 25% increase cutoff used in the manuscript.

From a clinical point of view, acquired resistance is usually described as an initially responding patient (CR, PR or SD >4 or 6 months) that later on presents PD to therapy. Were the 7 pts with ARID1A mutation responders to treatment? Did they discontinue treatment because of PD or due to other cause (secondary effects, resectable disease)? Authors should describe specific clinical data on the 7 pts with emergence of ARID1A mutation to better understand whether emergence of ARID1A reflects clinical acquired resistance. Would also be relevant to include information on scheme of chemotherapy. Impact of chemotherapy in ARID1A related resistance should be better characterized: is baseline and enrichment of ARID1A in cetuximab vs bevacizumab independent of FOLFOX vs FOLFIRI arm?

Regarding primary resistance of ARID1A in the CALGB 80405 study, the numbers are very low in the OS analysis (17 pts ARID1A mutant treated with cetuximab) and therefore results should be interpreted with caution. Results on PFS regarding ARID1A and treatment arm are not statistically significant, again the numbers are very low. Again, results should be adjusted by scheme of chemotherapy (FOLFOX vs FOLFIRI). Similar low numbers for OS analysis on real-world data, with no statistically significant results for OS and treatment, and no data on PFS.

Methods Section

More details are needed on cfDNA methodology: details on extraction timepoints (i.e., was there a time window for the end of treatment sample extraction, this may influence the detection of mechanisms of resistance) which tubes were used for collection?, median ml of plasma per sample?, protocol for DNA extraction, DNA quality controls,

Figure 3 legend is confusing. In Fig 3d and 3f, PD-mild, SP-PD, SD-PR should be described.

Reviewer #4:

Remarks to the Author:

The study by Johnson and colleagues presents an interesting perspective on ARID1A alterations and their resistance to anti-EGF drugs. This finding stems from the exploration of cfDNA of patients treated with this drug. Subsequently, they further investigate this potential novel resistance mechanism by using four! different colorectal datasets that include DNA mutation data, RNA expression profiling and drug sensitivity data. However, we think that this paper requires more validation to strongly support their conclusion. In general, my concerns are focused on mutation calling, assigning mutation pathogenicity estimates (to ARID1A) genes, confounding factors, the lack of positive control genes and benchmarking the performance of the ARID1A expression signature. Also, most sections of the text

are too dense, not well structured, and the reader can be flooded with too many details. For instance, Fig 2 b is discussed in a later section of the results. Lastly, the paper reference list is not numbered making it impossible to review the paper citations.

1. mutation calling

“Consistent with the literature, we observed reduced circulating tumour DNA (ctDNA) upon chemotherapy treatment, with a subsequent increase at progression.” Indeed, Fig 1a shows a ~50% drop in mutation prevalence from untreated (PRE) to chemo-treated (OTH) cf-DNA and also EOT samples show an overall lower detection limit than PRE samples. The authors should show that the here observed effect of ARID1A cannot be explained by missing mutations of cfDNA mutation detection. Have the author’s sequenced independent samples of the same donor to get insight into the sensitivity of cfDNA mutation calling, as well as in the observation of similar allele frequencies. My main concern is that EGFR mutations upon anti-EGFR treatment is a well-described resistance mechanism, and EGFR mutation enrichment (neither in numbers nor in allele frequency) is not observed in this analysis, which questions how reliable this mutation dataset can be used for enrichment analysis.

2. mutation pathogenicity

The authors should discuss the mutation types of ARID1A mutations. ARID1A is a tumour suppressor gene and inactivation by a simple mutation evens must include inactivating nonsense or out of frame indel mutation or rearrangement mutation. This information can be added in Fig 1. D. Moreover, information on pathogenicity should be explored consistently in every dataset that has been used to detect samples with ARID1A mutations. Sometimes, the authors refer to non-synonymous mutations, for other datasets they used SELECT to determine known or likely oncogenic significance and in the cfDNA dataset they consider point mutations (also missense?), indels and rearrangements.

3. confounding factors

As the authors explained in the text: “intrinsic ARID1A mutations in CRC have commonly been associated with microsatellite instable-high (results section 2) and BRAF-mutations (results section 3)”. BRAF is one of the key components in the EGFR/MAPK signalling pathway. Therefore, the authors have corrected for KRAS, NRAS and BRAF mutations when exploring the FH-FMI CGDB database. However, the authors must also explore the co-occurrence of these mutations in every dataset, including the data of the cfDNA. Nearly 20% of the samples carry a BRAF mutation before treatment, which is twice as many ARID1A mutations.

Also, ARID1A is a very long gene which likely explains why MSI-H samples often acquire ARID1A mutations. These passenger mutations likely have no pathogenic impact. Therefore, the authors must explore the pathogenicity of ARID1A mutations (see point 2), and include MSI status in every dataset (e.g. MSI status is reported for TCGA dataset) if possible.

4. lack of positive control genes

As the authors have discussed, EGFR mutations (and also other EGFR/MAPK pathway genes) play an important role in the resistance to anti-EGFR therapy. These genes can thus not only be used as a positive control in every dataset but also to show how the effect of EGFR mutations (mutation prevalence / clinical outcome) relates to ARID1A mutations.

5. benchmarking the performance of the ARID1A expression signature.

The expression signature tool is not thoroughly described, neither correctly benchmarked. The authors must explore the performance of the ARID1A mutant-like signature score. I.e. the number of false-positive / negative in test and control group of ARID1A mutant and WT samples. Using the signature, only 30 ARID1Amut samples are identified in CALGB/SWOG dataset whereas 41 samples actually carry an ARID1A mutation.

Also, do the authors find the same mutations differentially expressed when they used the samples of the CALGB/SWOG dataset, and not from the TCGA? Lastly, the authors state that ARID1A “likely

requiring biallelic loss to confer resistance". Why is ARID1A expression then not observed as significantly lower expressed in ARID1A mutations as compared to ARID1A wild type samples?

6. In vitro validation

The authors found a correlation relationship between ARID1A and anti-EGFR resistance, but no causation. The use of ARID1A manipulated human cell systems may unambiguously show a causal relationship, but these experiments require wet-lab resources and activities. As an alternative, the authors could explore cancer cell line models treated with anti-EGFR in well-controlled in vitro assays. The GDSC database of Sanger (<https://www.cancerrxgene.org>) and the Cancer Cell Line Encyclopedia (<https://www.nature.com/articles/s41586-019-1186-3>) can be used to link genomic profiles with IC50 values of anti-cancer drugs, including anti-EGFR.

Minor comments

The authors have mined several databases to identify and split up samples in ARID1A mutations and wild type categories. Unfortunately, sample information is absent in the methods section although this information is crucial to independently validate their findings. Therefore, the authors should make an overview of the used samples for each dataset and which mutations are considered as pathogenic ARID1A mutations. For instance, sample counts for the FH-FMI CGDB database are completely missing.

I do not understand fig 1 A, legend says it shows the number of patients, but y axis shows the number of mutations and bars represent a fraction. Also, can the authors explain the 36% in cetuximab and 5% in bevacizumab. And what is the difference between a selected alteration and not selected alteration? This is also important to understand suppl fig 1b, where a large difference in the patient count has been found for ARID1A mutations. After carefully exploring Suppl fig 1a, one could not draw the conclusion that ARID1A mutations differ between Cetux and Bev, or between PRE and EOT.

For consistency with CALBG80405, ... What is CALBG80405? It has not been described in the results section, neither in the methods section.

A worse survival response of cancer patients with ARID1A mutations has been analysed in an independent dataset of FH-FMI CGDB database. However, there is no

Fig 2 E shows indeed a significant effect between mutant-like and wt-like. Can the authors comment on why the mutant-like group shows an increase in signature score whereas the discovery and validation cohort (fig 2 C) show a decrease?

The authors have used the TCGA dataset to construct ARID1A expression signature, however, the TCGA data contains also clinical outcome and can thus be explored as an independent dataset as well.

In supp fig 5, the authors explore the ARID1A expression signature for genes involved in the eRAS/BRAF pathway and observe for 4 genes (for 3 genes a significant effect) a lower signature score. Can the authors perform this analysis also to genes not active in the BRAF pathway? Ideally, the authors can apply this approach to every gene and see if the eRAS/BRAF related genes are top-ranked.

The authors applied the ARID1A signature to 578 donors of CALGB/SWOG dataset and found 30 donors with ARIDmut and 267 ARIDwt. What about the other 281 samples?

Point by point response to reviewers:

Reviewer #1 (Remarks to the Author): Expert in CRC genomics

The manuscript by Johnson et al. addressed longitudinal CRC cohorts treated with either cetuximab or bevacizumab, investigating mutations in cfDNA using the FM cancer panel. They show that a fraction of the patients that develop resistance to Cetuximab treatment display selected mutations in ARID1A, and that patients carrying intrinsic ARID1A mutations treated with cetixumab have a lower OS than the ones who received bevacizumab. The association of ARID1A mutations with treatment response in CRC is interesting and has important implications for the patient management. The authors report mutually exclusive genomic alterations in EGFR and ARID1A, however previous publications (e.g Karachaliou et al. 2018) mentioned that those could co-occur in lung carcinoma for instance.

We appreciate the reviewer's comments. Thank you for pointing us to this article. Karachaliou et al. 2018, J Thorac Oncol. reported observing 5/23 co-occurring ARID1A and EGFR mutations (EGFR L858R and EGFR del19) in tumors from NSCLC patients. We have also observed a few sporadic co-mutated cases, however, based on our analysis, these co-mutation events should be quite rare. Although we can't fully explain the apparent co-mutation frequency discrepancy, it is important to note that the analysis by Karachaliou et al was based on NGS data from only 23 liquid biopsy samples. In contrast, our analysis was based on the 29,757 NSCLC sequenced tumor tissue genomes with a CLIA validated methodology (FMI panel). Since the SELECT algorithm was run adding MSI status as a covariate, we have now performed the fisher-exact test in the MSS and MSI-H cohort separately (see data below, Reviewer Figure 1) and obtained consistent results. Furthermore, in the FMI core NSCLC data-set, ARID1A and EGFR co-mutations were observed only in 100 tumors out of 1,727 ARID1A mutants and 4,153 EGFR mutants out of the out of 29,757 lung tumors evaluated (Figure 4C). Consistently with our data. Hung et al., 2020¹ have shown that ARID1A mutation and aberrant expression by IHC also correlate with a lack of EGFR mutations.

Reviewer Figure 1. Overview of genes with SV or copy number alterations (CN) that are mutually exclusive (ME, purple) or co-occurring (CO, green) with ARID1A SVs based on the SELECT method² (Mina et al., 2017), in a CRC (n = 16 931) from the FMI FoundationCORE database³. Known and likely oncogenic alterations in each gene were considered for the SELECT analysis. Asterisks indicate significant relationships (FDR < 0.1).

The second part of the manuscript regarding the EGFR-like expression signature is hard to follow and must be clarified. Besides, the methods pertaining to RNAseq are very short and lack important information.

Thank you for the feedback. We apologize for the confusing text. In an attempt to address this concern, we updated the text accordingly in the results and methods section to explain in further detail how the ARID1A signature score was developed. Briefly, the ARID1A mutant-like signature was derived from the list of differentially expressed genes (DEG) between ARID1A mutant vs WT colon tumors from the TCGA cohort. To enrich for ARID1A mutations that would impact the function of the protein in TCGA, ARID1A mutant tumors were defined as those with ARID1A somatic mutations annotated as selected functional events (SFE) in Mina et al., 2017² for TCGA colon cancer tumors. These alterations were curated from the thousands of observed copy-number alterations (CNAs) and somatic mutations reported in the TCGA. The ARID1A mutant-like signature score was calculated from the PC1 scores of the principal component analysis of gene expression matrix corresponding to this DEGs for the TCGA and CALGB 80405 cohort. To enrich ARID1A mutant tumors, we considered tumors with an ARID1A signature score in top quartile (Q4) as ARID1A mutant-like and those with a score in the other quartiles (Q1-Q3) as ARID1A WT-like.

To examine the relationship between the ARID1A mutant-like signature and existing oncogenic signatures, we calculated a per sample (ssGSVA, Hänzelmann et al, 2013⁴) score and looked for correlation between the ARID1A mutant-like signature scores with the ssGSVA scores calculated for each C6 oncogenic pathway. The top correlated pathways with $R > 0.4$ and $p < 0.0001$ included the EGFR_UP.V1_UP oncogenic signature. Since the EGFR_UP.V1_UP oncogenic signature corresponds to

the genes up-regulated in MCF-7 cells engineered to express ligand-activatable EGFR, we hypothesized that ARID1A mutants might enable resistance to anti-EGFR by activating a ligand-independent activation of downstream EGFR transcriptional changes.

The accompanying Supplementary Table does not show directly gene expression values (minor remark, it displays 61 genes and not 64 as indicated in the text), nor the range of expression differences.

Thank you for bringing this unintended but important oversight to our attention. The text has been corrected and the raw data for the RNAseq data has been deposited at GEO (Accession # GSE196576). The per sample gene enrichment scores and ARID1A mutant-like signature group has been added to the CALGB metadata and added as supplemental data (CALGB_metadata_arid1ms_Nov16_2021.xlsx).

It would be interesting to annotate the samples according to the different CRC tumor groups identified in previous studies, which might be relevant in the data interpretation. It is also unclear whether Folfox or FOLFIRI have a differential impact on the gene signatures. In the PDX section, the authors mention that the genetic information was not available, but it should be possible to extract it for the RNAseq data. Though there are indications of a potential connection between ARID1A mutations and the EGFR pathway, this is not clearly demonstrated in the present form of the manuscript. This study is interesting but could be strengthened.

Thank you for your suggestions. We have added supplemental Fig. 8 & 9 to report the prevalence of ARID1A WT and mutant tumors by MSI status, CMS subtype annotation using CMScaller⁵ and supplemental Fig. 10 for chemotherapy protocol (FOLFOX vs FOLFIRI). We also explored the relationship between CMS subtype, MSI status and chemotherapy protocol with the ARID1A mutant-like signature score in both CALGB and TCGA CRC cohorts.

Supplemental Fig. 8. ARID1A mutant-like signature scores by MSI status in TCGA (a) and CALGB/SWOG 80405 (b). The proportion of ARID1A WT and mutant tumors in MSI and MSS tumors (*left*) and ARID1A mutant-like signature scores stratified by ARID1A mutation and MSI status (*right*).

Supplemental Fig. 9. ARID1A mutant-like signature scores by CMS subtype in TCGA (a) and CALGB/SWOG 80405 (b). Heatmap shows results from CMScaller mRNA gene set analysis, confirming enrichment of known characteristics in each CMS group (left). The proportion of ARID1A WT and mutant tumors in MSI and MSS tumors (middle) and ARID1A mutant-like signature scores stratified by ARID1A mutation and MSI status (right).

Supplemental Fig. 10. ARID1A mutant-like signature scores by chemo protocol in CALGB/SWOG 80405. The proportion of ARID1A WT and mutant tumors in FOLFOX and FOLFIRI treated groups (*left*) and ARID1A mutant-like signature scores stratified by chemo protocol used (*right*).

In the PDX section, only microarray gene expression data (GEO accession # GSE76402) is available for the published cohort so we were unable to extract genetic information from these samples. To strengthen the connection between ARID1A and EGFR we have revisited the text to highlight the correlation between ARID1A mutant like signature and EGFR oncogenic signature from Molecular Signatures Database (MSigDb) and the mutual exclusivity of ARID1A and EGFR mutations in lung tumors. We also cite Liao et al., 2017⁶, *Genes & Dev* earlier in the paper to highlight the cell line crisper and shRNA functional genetic cell screen data with anti-EGFR inhibitor gefitinib that first showed ARID1A as a potential mechanism of resistance to EGFR targeted therapies in lung cancer. This previously published data is consistent with our findings in patients and provides further biological evidence of the ability of ARID1A loss to provide a mechanism of resistance to EGFR signaling inhibition.

Reviewer #2 (Remarks to the Author): Expert in ARID1A

Johnson et al. perform targeted sequencing of cell-free DNA from colorectal cancer patients with drug resistant disease. They show that ARID1A mutations are enriched in emerging mutations from CRC patients treated with anti-EGFR (cetuximab) therapy. Although aspects of the research are not new, including the connections between ARID1A and EGFR/MAPK signaling, the work is a tour de force with high clinical and translational impact. In addition, the rigor of the study is strong and the results are conclusive. Therefore, I support publication.

Minor comments:

-ARID1A mutations are associated with tumor progression, chemoresistance and metastatic spread in cancer, and associations between ARID1A and MAPK in colorectal cancer have been reported. There should be more discussion on what's known in this regard. In addition, strong statements, such as, 'a connection between ARID1A mutations and EGFR/MAPK has not been previously reported,' should be omitted from the abstract, since there are published reports on this subject.

Thank you for your encouragement and positive feedback. We have taken out the statement "a connection between ARID1A mutations and EGFR/MAPK has not been previously reported". In response to your input, we cited additional papers reporting an association between ARID1A and MAPK in colorectal cancer and added text and references to further elaborate on the known role of ARID1A mutations in tumor progression, chemoresistance and metastatic spread in cancer.

Reviewer #3 (Remarks to the Author): Clinical CRC expert

The manuscript by Johnson et al identifies ARID1A as a mechanism of acquired resistance to antiEGFR based-therapy. Moreover, it describes ARID1A as a marker of primary resistance to antiEGFR therapy. The work is original and novel. However, although the manuscript provides evidence based on analysis of data from a randomized clinical trial and validation cohorts (including PDX and real-world data), the unfrequent percentage of the event (ARID1A mutation) limits the clinical interpretation and results should be cautiously interpreted. Moreover, because cetuximab/panitumumab/ bevacizumab are administered together with chemotherapy, the impact of the chemotherapy scheme (irinotecan vs oxaliplatin) may be a confounding factor. To add robustness to the causality of ARID1A in driving acquired and primary resistance to anti-EGFR therapy, functional studies in preclinical models (i.e., ARID1A overexpression confers resistance?, ARID1A knockdown reverts resistance?) are recommended.

Thanks for your feedback and recommendations, we agree that the prevalence of ARID1A mutations is low in CRC (11%, <https://www.cbioportal.org/>) and that acknowledge the limitations of the correlative data given the small N. In response to your important point about possible confounder effects of chemo backbones, we now assessed the impact of the irinotecan vs oxaliplatin on the ARID1A mutant-like signature score and tested the effect of the chemo protocol regimen FOLFOX vs FOLFIRI in our coxph models. When we evaluated the effect of chemo protocol and arm for OS ($p = 0.9$) and PFS ($p = 0.9$), these were not significant. We also added chemo protocol as a covariate in our multivariable model on the eRAS/BRAF WT cohort and observed similar results with the interaction term between the ARID1A mutant-like signature and arm B (cetuximab) being significant for OS (HR: 5(1.9-13); $p=0.001$) and PFS (HR: 3.4(1.4-8.3); $p=0.007$).

OS (see below)

Hazard ratio

PFS (see below)

Hazard ratio

We also observed similar results as those we reported in the paper with the interaction term between ARID1A mutation status and arm B (cetuximab) being significant for OS (HR: 19(4.6-79); p=5e-05) but not PFS (HR: 2.8(0.8-10);p=0.1).

OS (see below)

Hazard ratio

PFS (see below)

Hazard ratio

We also repeated the analysis adding chemo protocol as a covariate in our multivariable model in the CGDB real-world data cohort, and consistent results with our primary analysis showed ARID1A mutations in eRAS/BRAF WT patients have worst outcome on cetuximab or panitumumab compared to those without ARID1A mutations.

Cetuximab/Panitumumab (KRAS/NRAS/BRAF WT): ARID1A Mut vs WT (Ref)

We have updated the manuscripts with forest plots with chemo protocol added as a variable (see supplemental Fig. 4, 5, 6b, 14, 15).

To further explore the hypothesis that ARID1A mutations confer resistance to anti-EGFR, we treated cetuximab sensitive cell line NCI-H508 with ARID1A knockdown sgRNA and showed increased pERK signaling and acquired resistance to cetuximab (Reviewer Figure 2, see below). These results are further supported by the results published by Liao et al.⁶, which showed ARID1A scored in the top 30 of both CRISPR and shRNA screens (combined FDR < 0.001) as mechanism overcoming resistance to EGFR signaling inhibition in NSCLC cell lines.

KO detected in all sgRNA except sg4
sg2 shows profound KO

All sgRNA but sg4 display elevated
MAPK signaling

CRISPR knockout of *ARID1A* in NCI-h508 CRC cell
line results in increased cetuximab resistance

Reviewer Figure 2. Effective CRISPR knockout of *ARID1A* in the cetuximab sensitive CRC cell line NCI-H508 confers elevated ERK phosphorylation and partial resistance to cetuximab in culture.

Specific comments to improve the interpretation of data and conclusions are:
ctDNA is a very appropriate tool to monitor for the emergence of clones of resistance, however interpretation of the results, relative MAFs and MAF's cutoffs are not currently standardized. The authors should clarify how the 25% increase in the relative MAF is calculated (i.e., relative to normal cfDNA in the sample? Relative to total ctDNA in the sample? relative to the highest MAF?). That may completely change the results. For example in Figure 1d, in the second evolution map it is not clear that *ARID1A* increase is not a consequence of an increase in total ctDNA (minor comment: please use same color for each specific mutation in all figures). Moreover, authors should justify the 25% increase cutoff used in the manuscript.

Thank you for raising these points. We apologize for the lack of clarity. We have clarified the text in the manuscript to address these concerns. The MAF was calculated relative to the highest reported MAF in the sample precisely because the total cfDNA is

variable between timepoints. By using the relative MAF we can compare the relative rank of the mutations between samples collected at different time points to look for genes with alterations whose relative rank has been increased post treatment to identify alterations selected for as a resistance mechanism. We looked at different thresholds and settled on 25% to prioritize genes and alterations enriched in post treatment samples. We also evaluated the distribution of the relative MAF changes in both arms in Fig. 1B to ensure we were seeing a shift in the distribution supporting preferential enrichment in the cetuximab treated group relative to the bevacizumab treated group.

For the evolution map, we apologize the color scheme caused some confusion between the MET R469P (in red) and ARID1A Exon 1, Q176fs (blood orange). We have added an arrow to highlight the band corresponding to the ARID1A mutation. To facilitate interpreting the results, we have also uploaded the raw and processed ctDNA data to dbGaP (<https://www.ncbi.nlm.nih.gov/gap/>, accession # TBD).

As you can see in the updated figure below the arrows show the absence of the ARID1A Exon 1, Q176fs at baseline and increased MAF at end of treatment.

Figure 1d. Evolution Map (arrow added).

From a clinical point of view, acquired resistance is usually described as an initially responding patient (CR, PR or SD >4 or 6 months) that to therapy. Were the 7 pts with ARID1A mutation responders to treatment? Did they discontinue treatment because of PD or due to other cause (secondary effects, resectable disease)? Authors should describe specific clinical data on the 7 pts with emergence of ARID1A mutation to better understand whether emergence of ARID1A reflects clinical acquired resistance. Would also be relevant to include information on scheme of chemotherapy. Impact of chemotherapy in ARID1A related resistance should be better characterized: is baseline and enrichment of ARID1A in cetuximab vs bevacizumab independent of FOLFOX vs FOLFIRI arm?

Thank you for your comments. To help address the points you raised, we have added a Supplemental table 2 that shows the specific clinical and sample collection data for the 7 patients with ARID1A emerging mutations.

	Treatment	Chemo_protocol	Best response	Progression date	PFS Status	PFS (days)	PRE	OTX	EOS	Days Bw PRE and OTX	Days bw PRE and EOS
SAANDA	Cetuximab	FOLFIRI	NA	6/25/07	Event	96	3/26/07	5/22/07	7/3/07	57	99
SAARSX	Cetuximab	FOLFOX	SD	8/23/10	Event	178	3/3/10	5/5/10	7/12/10	63	131
SAASGM	Cetuximab	FOLFIRI	PR	10/2/13	Event	813	7/13/11	9/7/11	10/25/12	56	470
SABCZU	Cetuximab	FOLFOX	SD	4/24/11	Event	153	11/30/10	NA	4/26/11	NA	147
SAANSJ	Cetuximab	FOLFOX	PR	9/16/08	Event	788	1/7/08	3/18/10	3/22/10	801	805
SAAAUP	Cetuximab	FOLFOX	PR	4/2/12	Event	333	5/5/11	7/14/11	11/3/11	70	182
SAAAKR	Bevacizumab	FOLFIRI	SD	6/6/12	Event	176	12/13/11	4/3/12	6/19/12	111	188

Supplemental Table 2. Abbreviations: PRE, at baseline; OTH, on treatment; EOT, at progression or end of study.

Unfortunately, given the small number of patients with emerging ARID1A mutations in the blood it is difficult to assess whether the enrichment of ARID1A mutations are independent of FOLFOX vs FOLFIRI arm. However, in our larger cohort analysis, we tested the effect of chemo protocol on the OS and PFS of patients with ARID1A mutations treated with cetuximab relative to bevacizumab and found no effect of chemo protocol on the interaction between ARID1A mutation status and worse OS when treated with cetuximab. We observed similar findings in our real world data cohort (see response to reviewer #1 above and supplemental Fig. 6b. We have added these points to the discussion of the paper.

Regarding primary resistance of ARID1A in the CALGB 80405 study, the numbers are very low in the OS analysis (17 pts ARID1A mutant treated with cetuximab) and therefore results should be interpreted with caution. Results on PFS regarding ARID1A and treatment arm are not statistically significant, again the numbers are very low. Again, results should be adjusted by scheme of chemotherapy (FOLFOX vs FOLFIRI). Similar low numbers for OS analysis on real- world data, with no statistically significant results for OS and treatment, and no data on PFS.

Thank you. We have revisited the text to reflect the caveat in the interpretation of these correlations due to the low numbers. We have also adjusted the scheme of chemotherapy and added the forest plots to the supplemental Fig. 4,5, 6b, 14, 15. When we adjusted for chemotherapy protocol, it did not change the results and remains significant suggesting that ARID1A mutations in eRAS/BRAF WT patients had worse outcome to cetuximab in two independent datasets albeit with low numbers in both studies.

Methods Section

More details are needed on cfDNA methodology: details on extraction timepoints (i.e., was there a time window for the end of treatment sample extraction, this may influence the detection of mechanisms of resistance) which tubes were used for collection?, median ml of plasma per sample?, protocol for DNA extraction, DNA quality controls,

We have added more details to the methods section to expand on the cfDNA methodology used in our study.

Figure 3 legend is confusing. In Fig 3d and 3f, PD-mild, SP-PD, SD-PR should be described.

The PDX data was taken from Isella et al, 2017⁷, Nat. Comm. study. The data plotted in 3c and 3f represent "Response to cetuximab: tumor volume variation after 3 weeks of treatment" and Cetuximab response was annotated as reported in Bertotti et al., 2011⁸. The values plotted on the y-axis are Variation of tumor growth from baseline, so we are looking at whether or not the cetuximab had an effect on tumor size after it has established a 400 mm³ average volume to minimize the growth rate effect.

From the methods section "Established tumors (average volume 400 mm³) were treated with the following regimens, either single-agent or in combination: cetuximab (Merck) 20 mg/kg, twice-weekly; [...] Tumor size was evaluated once weekly by caliper measurements and the approximate volume of the mass was calculated using the formula $4/3\pi \cdot (d/2)^2 \cdot D/2$, where d is the minor tumor axis and D is the major tumor axis."

The RECIST-like cetuximab response categories were defined as follows for Fig. 3d and 3f from the Bertotti et al., 2011⁸ paper.

Figure 2. Effect of cetuximab treatment in unselected metastatic colorectal cancer xenopatiens. **A**, waterfall plot of cetuximab response after 3 weeks of treatment, compared with tumor volume at baseline, in an unselected population of 47 cases. Dotted lines indicate the cut-off values for arbitrarily defined categories of therapy response: cases experiencing disease progression, stabilization, or regression are shaded in light brown, light yellow, and light aquamarine, respectively. Asterisks denote the samples for which growth curves are shown in panel B. **B**, representative tumor growth curves in cohorts derived from individual patients, treated with placebo (gray) or cetuximab (red). $n = 6$ for each treatment arm. R, regression/shrinkage; S, stabilization/disease control; S/P, initial stabilization followed by tumor progression; P, progression/lack of response.

Reviewer #4 (Remarks to the Author): Expert in mutational signatures
The study by Johnson and colleagues presents an interesting perspective on ARID1A alterations and their resistance to anti-EGF drugs. This finding stems from the

exploration of cfDNA of patients treated with this drug. Subsequently, they further investigate this potential novel resistance mechanism by using four! different colorectal datasets that include DNA mutation data, RNA expression profiling and drug sensitivity data. However, we think that this paper requires more validation to strongly support their conclusion. In general, my concerns are focused on mutation calling, assigning mutation pathogenicity estimates (to ARID1A) genes, confounding factors, the lack of positive control genes and benchmarking the performance of the ARID1A expression signature. Also, most sections of the text are too dense, not well structured, and the reader can be flooded with too many details. For instance, Fig 2 b is discussed in a later section of the results. Lastly, the paper reference list is not numbered making it impossible to review the paper citations.

We apologize for the formatting issues with the manuscript and references. We revised the text to address these issues in the updated version of this manuscript.

1. mutation calling “Consistent with the literature, we observed reduced circulating tumour DNA (ctDNA) upon chemotherapy treatment, with a subsequent increase at progression.” Indeed, Fig 1a shows a ~50% drop in mutation prevalence from untreated (PRE) to chemo-treated (OTH) cf-DNA and also EOT samples show an overall lower detection limit than PRE samples. The authors should show that the here observed effect of ARID1A cannot be explained by missing mutations of cfDNA mutation detection. Have the author’s sequenced independent samples of the same donor to get insight into the sensitivity of cfDNA mutation calling, as well as in the observation of similar allele frequencies. My main concern is that EGFR mutations upon anti-EGFR treatment is a well-described resistance mechanism, and EGFR mutation enrichment (neither in numbers nor in allele frequency) is not observed in this analysis, which questions how reliable this mutation dataset can be used for enrichment analysis.

Thank you for your feedback. We completely agree that it was unexpected to not find EGFR extracellular domain (ECD) mutations. However, we did find KRAS mutations which are also a prevalent and well-established mechanism of resistance to cetuximab. Although the reason for the lack of detection of EGFR ECD emerging mutations in this data-set is presently unknown, it is tempting to speculate that these acquired mutations are perhaps less prevalent in the first line setting. It has been reported that, in circulating cell-free tumor DNA of patients treated with anti-EGFR antibodies, RAS mutations emerge earlier than EGFR ECD variants. Furthermore, subclonal RAS but not EGFR ECD mutations are present in CRC samples obtained before exposure to EGFR blockade⁹, thus suggesting that perhaps EGFR-ECD could represent “*de novo*” mutations. Thus, there is the possibility that the duration of cetuximab treatment and time of sample collection in this 1L study was not long enough to see the emergence of EGFR ECD mutations. In our cohort, we identified 4/19 and 3/13 patients with acquired EGFR mutations in the bevacizumab and cetuximab arm, respectively (Supplemental Table 1 and Reviewer Figure. 3, see below). However, we did not find preferential

enrichment of EGFR mutations in the cetuximab treated group relative to the bevacizumab group (Fisher exact test p value = 0.61).

Reviewer Figure 3. EGFR mutations detected in cfDNA from CALGB/SWOG 80405 trial liquid biopsies. (a) Distribution of the % change in relative MAF between baseline and end of study for all EGFR alterations detected in all patients. Dashed line represents 25% threshold used to call tumors with selected EGFR mutations. (c) Clinical information for patients with selected ARID1A, KRAS, and/or EGFR alterations.

2. mutation pathogenicity

The authors should discuss the mutation types of ARID1A mutations. ARID1A is a tumour suppressor gene and inactivation by a simple mutation events must include inactivating nonsense or out of frame indel mutation or rearrangement mutation. This information can be added in Fig 1. D. Moreover, information on pathogenicity should be explored consistently in every dataset that has been used to detect samples with

ARID1A mutations. Sometimes, the authors refer to non-synonymous mutations, for other datasets they used SELECT to determine known or likely oncogenic significance and in the cfDNA dataset they consider point mutations (also missense?), indels and rearrangements.

Thank you for your suggestions. Unfortunately, it was not possible to standardize the annotations used to call ARID1A mutations across platforms due to proprietary pipelines used by respective vendors to annotate variants. When possible, we used the annotations most likely to enrich for mutations with known pathogenicity or likely functional impact on the protein. For example, for Foundation Medicine (FMI) assays we restricted our analysis to the known and likely variants annotated by FMI. For TCGA, we enriched for alterations likely to impact the function of the protein by using the curated annotation from Mina et al., 2017². Lastly, for the Guardant cfDNA assay we restricted the analysis to non-synonymous variants and evaluated the impact of the mutations post-hoc. In Fig. 1, 4/6 patients with acquired ARID1A mutations had frameshift mutations (A2240fs, Q176fs, P224fs, D1850fs) and 2/6 with a missense mutation (P1042L and H842R). Although H842R missense mutations have not been reported before, we did find two patients in the Foundation Core database with frameshift mutations affecting H842 (H842Afs*30, and H842Dfs*16).

3. confounding factors

As the authors explained in the text: "intrinsic ARID1A mutations in CRC have commonly been associated with microsatellite instable-high (results section 2) and BRAF-mutations (results section 3)". BRAF is one of the key components in the EGFR/MAPK signalling pathway. Therefore, the authors have corrected for KRAS, NRAS and BRAF mutations when exploring the FH-FMI CGDB database. However, the authors must also explore the co-occurrence of these mutations in every dataset, including the data of the cfDNA. Nearly 20% of the samples carry a BRAF mutation before treatment, which is twice as many ARID1A mutations. Also, ARID1A is a very long gene which likely explains why MSI-H samples often acquire ARID1A mutations. These passenger mutations likely have no pathogenic impact. Therefore, the authors must explore the pathogenicity of ARID1A mutations (see point 2), and include MSI status in every dataset (e.g. MSI status is reported for TCGA dataset) if possible.

Thank you for your comments. For the discovery analysis in Fig.1 evaluating ctDNA we considered all samples and looked post-hoc of the mutations co-occurred with known resistance mechanisms including KRAS/NRAS/BRAF (Fig. 1c, supplemental table 1) and copy number changes reported in Misale et al. 2014¹⁰, *Cancer Discovery* in Reviewer Figure 4 (see below). As reported in Fig.1C with the exception of 1/6 patients with a concurrent KRAS mutations and predicted to be MSI using Guardant's pipeline. The remaining 5/6 samples did not have a co-occurring mutation or copy number amplification associated with resistance in their end of treatment liquid biopsy, suggesting ARID1A mutations may be an additional mechanism of promoting resistance to cetuximab in these patients.

Patients with acquired ARID1A mutations in ctDNA samples

Bevacizumab-treated

Cetuximab-treated

Reviewer Figure 4. Non-synonymous SNV and CN mutations detected in cfDNA from patients with acquired ARID1A mutations.

MSI status was added as a covariate in all analyses. KRAS/NRAS/BRAF mutation status was added as a covariate in our differential gene expression analysis to generate the ARID1A mutant-like signature. We added supplemental Fig. 8 to show the relationship between the ARID1A mutant-like signature with MSI status. For all clinical survival analysis, MSI status was added as a covariate and forest plots have been added to supplemental Fig. 4, 5, 6b, 14, 15. The only exception was for the PDX cohort from Isella et al., 2017 for which no MSI status was available. However, they did profile the mutation status of KRAS, BRAF and NRAS, and hence our analysis focused on eRAS/BRAF WT PDX tumors only.

4. lack of positive control genes

As the authors have discussed, EGFR mutations (and also other EGFR/MAPK pathway genes) play an important role in the resistance to anti-EGFR therapy. These genes can thus not only be used as a positive control in every dataset but also to show how the effect of EGFR mutations (mutation prevalence / clinical outcome) relates to ARID1A mutations.

Pathogenic EGFR mutations are low 0.4% in CRC at diagnosis with 2/505 colorectal tumors patients with SFE in TCGA, Mina et al., 2017²). We observed a similar prevalence in our treatment -naïve first-line metastatic colorectal cohort CALGB/SWOG 80405 at 0.5% with 3/562 patients FMI tissue known/likely mutation calls. Given the very low prevalence of EGFR mutations at baseline in CRC, this is not an appropriate control. Instead, we included KRAS as a positive control since it is a well-established mechanism of resistance to anti-EGFR therapy. In contrast EGFR mutations are more prevalent in lung tumors, and consistent with our hypothesis we found EGFR known and likely pathogenic mutations are mutually exclusive with ARID1A mutations in lung tumors (Fig 4).

5. benchmarking the performance of the ARID1A expression signature. The expression signature tool is not thoroughly described, neither correctly benchmarked. The authors must explore the performance of the ARID1A mutant-like signature score. I.e. the number of false-positive / negative in test and control group of ARID1A mutant and WT samples. Using the signature, only 30 ARID1A mut samples are identified in CALGB/SWOG dataset whereas 41 samples actually carry an ARID1A mutation.

We apologize for the confusion, only 30/41 patients with ARID1A mutation status had RNAseq available to evaluate the relationship between ARID1A mutation status and ARID1A mutant-like signature score.

Also, do the authors find the same mutations differentially expressed when they used the samples of the CALGB/SWOG dataset, and not from the TCGA?

The differentially expressed genes from the CALGB/SWOG dataset and the signature derived from these genes produced similar results.

Lastly, the authors state that ARID1A “likely requiring biallelic loss to confer resistance”. Why is ARID1A expression then not observed as significantly lower expressed in ARID1A mutations as compared to ARID1A wild type samples?

Thanks for your comments. Our ARID1A signature was meant to enrich for tumors that carry ARID1A mutation or defects consistent with a loss of ARID1A function such as impaired SNF5 complex activity. As shown in Fig. 2d and Supplemental Fig. 7a, the ARID1A mutants are enriched in the 4th quartile of the ARID1A mutant-like signature score and hence, tumors were stratified as ARID1A mutant-like tumors when they fell in the 4th quartile and WT-like if their score fell in the other quartiles. To support using the ARID1A mutant-like signature to enrich for ARID1A mutations with a loss of function phenotype, we determined whether the ARID1A mutation score was enriched in tumors with mutations in tumor suppressor genes affecting the SWI/SNF complex and SNF5 activity. Consistent with our hypothesis, we found increased ARID1A mutant-like signature scores in tumors with selected functional mutations (as defined by Mina et al., 2017²) in other SWI/SNF complex genes including ARID2, PBM1, SMARCA4, and ARID1B (Supplemental Fig. 11). There was also a positive correlation between our ARID1A mutant-like signature score and the C6 oncogenic pathway derived from genes up-regulated in MEF cells (embryonic fibroblasts) with knockout of Snf5. Snf5 (Ini1/Baf47/Smrca1), a core member of the Swi/Snf chromatin remodeling complex and a potent tumor suppressor. Biallelic loss of Snf5 leads to the onset of aggressive cancers in both humans and mice. Since our ARID1A mutant-like signature is correlated with loss of SNF5 activity, we hypothesize biallelic loss of SWI/SNF activity may be necessary to confer resistance. However, we don't necessarily expect ARID1A biallelic loss to confer resistance and have corrected the text to clarify that point.

6. In vitro validation

The authors found a correlation relationship between ARID1A and anti-EGFR resistance, but no causation. The use of ARID1A manipulated human cell systems may unambiguously show a causal relationship, but these experiments require wet-lab resources and activities. As an alternative, the authors could explore cancer cell line models treated with anti-EGFR in well-controlled in vitro assays. The GDSC database of Sanger (<https://www.cancerrxgene.org>) and the Cancer Cell Line Encyclopedia (<https://www.nature.com/articles/s41586-019-1186-3>) can be used to link genomic profiles with IC50 values of anti-cancer drugs, including anti-EGFR.

Thank you for your suggestion. We have looked into both the DEPMAP and the Sanger GDSC database and added the results below. We restricted our analysis to lung adenocarcinoma cell lines and Osimertinib, 3rd generation anti-EGFR inhibitor since most colorectal cell lines are resistant to anti-EGFR targeted therapies at baseline. In fact, we could only find one colorectal cell line that was sensitive to cetuximab to perform our preliminary validation experiments (see response to reviewer #3). Unfortunately, the cell line numbers are low but we do see a trend for higher IC50 values for ARID1A mutant lung cancer cell lines compared to ARID1A WT cell lines treated with Osimertinib (Reviewer Figure 5 & 6, see below).

Reviewer Figure 5. Osimertinib cell line drug sensitivity data in DEPMAP.

Reviewer Figure 6. Osimertinib cell line drug sensitivity data in GDSC database.

Also as mentioned above the best causal evidence are provided by the functional genetic screen for EGFR inhibitors resistant mechanisms published by Liao et al., which showed ARID1A scored in the top 30 of both CRISPR and shRNA screens (combined FDR < 0.001) as mechanism able to overcome resistance to EGFR signaling inhibition in NSCLC cell lines.

Minor comments

The authors have mined several databases to identify and split up samples in ARID1A mutations and wild type categories. Unfortunately, sample information is absent in the methods section although this information is crucial to independently validate their findings. Therefore, the authors should make an overview of the used samples for each dataset and which mutations are considered as pathogenic ARID1A mutations. For instance, sample counts for the FH-FMI CGDB database are completely missing.

Thank you for your suggestions. We have provided more details on where the data can be downloaded for all cohorts in the "Data availability section" of the manuscript including the link to the SFE mutation annotations used from the Mina et al., 2017² study. For all Foundation Medicine data, variants annotated as "known" or "likely" were considered mutant. An overview of the somatic variants annotated as mutant is shown in Figure 4D. All ctDNA data including Guardant pipeline mutation annotations for CALGB/SWOG 80405 have been deposited to dbGaP (<https://www.ncbi.nlm.nih.gov/gap/>, accession # TBD).

I do not understand fig 1 A, legend says it shows the number of patients, but y axis shows the number of mutations and bars represent a fraction. Also, can the authors explain the 36% in cetuximab and 5% in bevacizumab. And what is the difference between a selected alteration and not selected alteration? This is also important to understand suppl fig 1b, where a large difference in the patient count has been found

for ARID1A mutations. After carefully exploring Suppl fig 1a, one could not draw the conclusion that ARID1A mutations differ between Cetux and Bev, or between PRE and EOT.

Figure 1A shows the number of patients with mutations detected for each gene on the x-axis layered above the red bar is the percentage of patients with selected alterations, defined as an increase in relative MAF greater than 25 percentage points between paired baseline and end of study cfDNA from the 333 1L mCRC patients treated with either bevacizumab or cetuximab in combination with chemotherapy in the CALGB/SWOG 80405. We have edited the figure legend to better convey this point. Supplemental figure 1B shows the overall prevalence % samples with mutations at each liquid biopsy collection. The data summarizes all samples sent for cfDNA sequencing not just the paired 333 ctDNA samples. To identify gene alterations that were enriched, we compared the normalized MAF to the highest reported MAF, between paired baseline and end of study cfDNA samples. We considered a mutation with an increase in MAF greater than 25 percentage points between end of treatment (EOT) and baseline (PRE) as being selected for by the treatment. We then compared the number of selected mutations in each gene between the treatment group using a two-tailed fisher exact test to determine if the selection of these mutations was specific to a specific arm. From this analysis, the only two genes that were statistically significant by Fisher-exact test (Supplemental Table 1) were KRAS and ARID1A (Supplemental Fig. 1B) although we found many selected mutations in a variety of genes including EGFR only ARID1A and KRAS were preferentially enriched in the cetuximab treated group relative to the bevacizumab.

For consistency with CALBG80405, ... What is CALBG80405? It has not been described in the results section, neither in the methods section.

We have added more details on the CALGB/SWOG 80405 to the result and methods sections.

A worse survival response of cancer patients with ARID1A mutations has been analysed in an independent dataset of FH- FMI CGDB database. However, there is no Fig 2 E shows indeed a significant effect between mutant-like and wt-like. Can the authors comment on why the mutant-like group shows an increase in signature score whereas the discovery and validation cohort (fig 2 C) show a decrease?

We apologize for the confusion we have replot the ARID1A mutant-like signature scores using a positive sign from the principal component analysis. Please see response to reviewer #1 for more details on how the signature was generated. We have also added more details on how the ARID1A mutant-like signature was generated to the methods section of the manuscript.

The authors have used the TCGA dataset to construct ARID1A expression signature, however, the TCGA data contains also clinical outcome and can thus be explored as an independent dataset as well.

Thank you for the suggestion. Unfortunately, TCGA doesn't have drug treatment information, which is what we need to assess whether TCGA metastatic CRC patients with ARID1A mutants do worse on anti-EGFR therapy compared to bevacizumab treated patients.

In supp fig 5, the authors explore the ARID1A expression signature for genes involved in the eRAS/BRAF pathway and observe for 4 genes (for 3 genes a significant effect) a lower signature score. Can the authors perform this analysis also to genes not active in the BRAF pathway? Ideally, the authors can apply this approach to every gene and see if the eRAS/BRAF related genes are top-ranked.

We apologize for the confusion. The genes *ARID2*, *PBRM1*, *SMARCA4*, and *ARID1B* shown in Supplemental Fig. 11 are part of the SWI/SNF complex. We restricted the analysis to other genes in the SWI/SNF complex to test the hypothesis that the ARID1A mutant-like signature enriches for loss of SWI/SNF complex activity. Since mutations in other genes part of the SWI/SNF complex have been shown to act as a tumor suppressor. We tested the effects of the 4 genes *ARID2*, *PBRM1*, *SMARCA4*, and *ARID1B* reported by Mina et al, 2017² with selected functional events.

The authors applied the ARID1A signature to 578 donors of CALGB/SWOG dataset and found 30 donors with ARIDmut and 267 ARIDwt. What about the other 281 samples?

RNAseq was run on 578 patient samples from the CALGB/SWOG 80405 cohort. However, the FMI tissue panel was only run on a subset of these samples and so we are left with 297 samples with both RNAseq and FMI mutation calls available. We have clarified the text to make this point clearer in the manuscript.

References:

- 1 Hung, Y. P., Redig, A., Hornick, J. L. & Sholl, L. M. ARID1A mutations and expression loss in non-small cell lung carcinomas: clinicopathologic and molecular analysis. *Mod Pathol* **33**, 2256-2268, doi:10.1038/s41379-020-0592-2 (2020).
- 2 Mina, M. *et al.* Conditional Selection of Genomic Alterations Dictates Cancer Evolution and Oncogenic Dependencies. *Cancer Cell* **32**, 155-168 e156, doi:10.1016/j.ccell.2017.06.010 (2017).
- 3 Hartmaier, R. J. *et al.* Genomic analysis of 63,220 tumors reveals insights into tumor uniqueness and targeted cancer immunotherapy strategies. *Genome Med* **9**, 16, doi:10.1186/s13073-017-0408-2 (2017).
- 4 Hanzelmann, S., Castelo, R. & Guinney, J. GSEA: gene set variation analysis for microarray and RNA-seq data. *BMC Bioinformatics* **14**, 7, doi:10.1186/1471-2105-14-7 (2013).
- 5 Eide, P. W., Bruun, J., Lothe, R. A. & Sveen, A. CMScaller: an R package for consensus molecular subtyping of colorectal cancer pre-clinical models. *Sci Rep* **7**, 16618, doi:10.1038/s41598-017-16747-x (2017).

- 6 Liao, S. *et al.* A genetic interaction analysis identifies cancer drivers that modify EGFR dependency. *Genes Dev* **31**, 184-196, doi:10.1101/gad.291948.116 (2017).
- 7 Isella, C. *et al.* Selective analysis of cancer-cell intrinsic transcriptional traits defines novel clinically relevant subtypes of colorectal cancer. *Nat Commun* **8**, 15107, doi:10.1038/ncomms15107 (2017).
- 8 Bertotti, A. *et al.* A molecularly annotated platform of patient-derived xenografts ("xenopatients") identifies HER2 as an effective therapeutic target in cetuximab-resistant colorectal cancer. *Cancer Discov* **1**, 508-523, doi:10.1158/2159-8290.CD-11-0109 (2011).
- 9 Van Emburgh, B. O. *et al.* Acquired RAS or EGFR mutations and duration of response to EGFR blockade in colorectal cancer. *Nat Commun* **7**, 13665, doi:10.1038/ncomms13665 (2016).
- 10 Misale, S., Di Nicolantonio, F., Sartore-Bianchi, A., Siena, S. & Bardelli, A. Resistance to anti-EGFR therapy in colorectal cancer: from heterogeneity to convergent evolution. *Cancer Discov* **4**, 1269-1280, doi:10.1158/2159-8290.CD-14-0462 (2014).

Reviewers' Comments:

Reviewer #4:

Remarks to the Author:

The authors modified the manuscript according to my previous suggestions.

Reviewer #5:

Remarks to the Author:

Main findings:

This manuscript by Johnson and colleagues reports that genetic or functional inactivation of the epigenetic regulator ARID1A associates with primary and acquired resistance to the EGFR antibody cetuximab in patients with metastatic colorectal cancer. The authors developed an ARID1A loss of function-like gene expression signature that not only identifies ARID1A mutant tumours but also more broadly other tumours with a potential impairment of the SWI/SNF complex. By applying this signature, ARID1A mutant-like tumours proved to be less responsive to anti-EGFR treatment than the ARID1A wild-type-like group in patients and in patient-derived xenografts. The ARID1A signature was enriched in transcriptional programmes consistent with high EGFR pathway activity and loss of SWI/SNF activity.

General assessment:

I have been invited to assess the extent to which the authors have responded to concerns raised by reviewers #1 and #3, who were unavailable to comment on this revised version of the manuscript. Replies to issues from reviewer #1 were adequate. However, I have strong reservations about how substantial issues from reviewer #3 were tackled. I concur with reviewer #3 that, owing to the rare frequency of ARID1A mutations, results should be cautiously interpreted and should be consolidated by gain- and loss-of-function mechanistic experiments in preclinical models. To address this request, the authors used CRISPR/Cas9 technology to knockout ARID1A in a cetuximab-sensitive colorectal cancer cell line and claim that ARID1A deletion led to increased ERK activation and reduced sensitivity to cetuximab (Reviewer Figure 2). In reality, results from this experiment are weak. Evidence was obtained in one cell line only; the effect size was negligible; there was no demonstration that ARID1A knockout resulted in the acquisition of the ARID1A loss of function-like gene expression signature. I assume that the authors themselves found this data set inconclusive, as they did not incorporate this piece of information in the revised manuscript. I strongly suggest the authors perform more functional experiments to provide their associations with causative significance. This is particularly critical not only to consolidate the biological relevance of their findings, but also because the study puts forward an interpretative conundrum. The authors state that ARID1A mutant/deficient, cetuximab-resistant tumours display higher levels of an EGFR gene expression signature than ARID1A wild-type tumours; yet, a body of evidence in preclinical models and in patients has documented that high EGFR pathway activity associates with responsiveness, rather than resistance, to EGFR antibodies in colorectal cancer (Khambata-Ford et al, *J. Clin. Oncol.* 25:3230-3237, 2007, PMID: 17664471; Jacobs et al., *J. Clin. Oncol.* 27:5068-5074, 2009, PMID: 19738126; Bertotti et al., *Nature* 526:263-267, 2015, PMID: 26416732; Isella et al., *Nat. Commun.* 8:15107, 2017, PMID: 28561063).

Point by point response to reviewers:

REVIEWERS' COMMENTS

Reviewer #4 (Remarks to the Author):

The authors modified the manuscript according to my previous suggestions.

Thank you!

Reviewer #5 (Remarks to the Author):

Main findings:

This manuscript by Johnson and colleagues reports that genetic or functional inactivation of the epigenetic regulator ARID1A associates with primary and acquired resistance to the EGFR antibody cetuximab in patients with metastatic colorectal cancer. The authors developed an ARID1A loss of function-like gene expression signature that not only identifies ARID1A mutant tumours but also more broadly other tumours with a potential impairment of the SWI/SNF complex. By applying this signature, ARID1A mutant-like tumours proved to be less responsive to anti-EGFR treatment than the ARID1A wild-type-like group in patients and in patient-derived xenografts. The ARID1A signature was enriched in transcriptional programmes consistent with high EGFR pathway activity and loss of SWI/SNF activity.

General assessment:

I have been invited to assess the extent to which the authors have responded to concerns raised by reviewers #1 and #3, who were unavailable to comment on this revised version of the manuscript. Replies to issues from reviewer #1 were adequate. However, I have strong reservations about how substantial issues from reviewer #3 were tackled. I concur with reviewer #3 that, owing to the rare frequency of ARID1A mutations, results should be cautiously interpreted and should be consolidated by gain- and loss-of-function mechanistic experiments in preclinical models. To address this request, the authors used CRISPR/Cas9 technology to knockout ARID1A in a cetuximab-sensitive colorectal cancer cell line and claim that ARID1A deletion led to increased ERK activation and reduced sensitivity to cetuximab (Reviewer Figure 2). In reality, results from this experiment are weak. Evidence was obtained in one cell line only; the effect size was negligible; there was no demonstration that ARID1A knockout resulted in the acquisition of the ARID1A loss of function-like gene expression signature. I assume that the authors themselves found this data set inconclusive, as they did not incorporate this piece of information in the revised manuscript. I strongly suggest the authors perform more functional experiments to provide their associations with causative significance. This is particularly critical not only to consolidate the biological relevance of their findings, but also because the study

puts forward an interpretative conundrum. The authors state that ARID1A mutant/deficient, cetuximab-resistant tumours display higher levels of an EGFR gene expression signature than ARID1A wild-type tumours; yet, a body of evidence in preclinical models and in patients has documented that high EGFR pathway activity associates with responsiveness, rather than resistance, to EGFR antibodies in colorectal cancer (Khambata-Ford et al, *J. Clin. Oncol.* 25:3230-3237, 2007, PMID: 17664471; Jacobs et al., *J. Clin. Oncol.* 27:5068-5074, 2009, PMID: 19738126; Bertotti et al., *Nature* 526:263-267, 2015, PMID: 26416732; Isella et al., *Nat. Commun.* 8:15107, 2017, PMID: 28561063).

Thank you for your feedback. We agree that the results from one cell line may not provide definitive evidence, which is why it was omitted from the manuscript and included as a reviewer only. The insights that our paper provided are based mainly in emerging mutations in patients under anti-EGFR treatment and in correlations between genetic and gene expression data and outcome also in patients. Experiments in Cell lines are beyond the scope of the present manuscript. However, consistently with our data in patients and PDX murine models, other authors have previously demonstrated in functional genetic screens that loss of function of ARID1A and other SWI/SNF components provide a resistance mechanism to EGFR inhibition in cell lines., Liao et al. (PMID: 28167502). Moreover Papadakis et al. 2015 (PMID: 25656847) showed that knockdown of ARID1A increased expression of EGFR transcript levels in two cell lines, further supporting a link between ARID1A deficiency and EGFR pathway.

Supplemental Figure 6 from Papadakis et al. 2015, Cell Reports

Figure legend: Multiple SWI/SNF components regulate EGFR expression and drug responses to MET and ALK inhibitors in NSCLC cells. (A) Downregulation of different SWI/SNF components all lead to upregulation of EGFR expression. SMARCE1, ARID1A, SMARCA4 and EGFR mRNA expression levels in the H1993 cells expressing pRS control or independent shRNA vectors targeting SMARCE1, ARID1A or SMARCA4 were measured by qRT-PCR. Error bars denote SD; * denote p values. [...] (D) Suppression of SMARCE1 or ARID1A but not SMARCA4 confers resistance to ALK inhibitors in EML4-ALK positive cells. H3122 cells described above (C) were grown in the absence or presence of 300 nM Crizotinib, 20 nM Ceritinib or 5 nM NVPTAE684. Cells were then fixed, stained and photographed after 10 days (untreated) or 28 days (treated). Taken from <https://www.nature.com/articles/cr201516> supplemental figure 6.

Lastly, efficacy preclinical data from Isella et al., Nat. Commun. 8:15107, 2017, PMID: 28561063 was used to support our observations in Figure 3 of the manuscript using the ARID1A mutant-like signature score in KRAS, NRAS (eRAS)/BRAF WT tumors showed that the tumors in the ARID1A mutant-like group are resistant to cetuximab (Figure 3c-f). Given the low overall frequency of ARID1A mutations in CRC tumors, it is possible these association were not reported previously. The Bertotti et al., Nature 2015, study evaluated 137 colorectal cancer patients, which may be the reason why the ARID1A mutations were not reported as a resistance mechanism to anti-EGFR therapy. In contrast, our study had 562 tumor biopsies and 333 paired liquid biopsies with baseline and end of treatment samples that were profiled for ARID1A and eRAS/BRAF mutation. Although KRAS WT ARID1A mutant patient transcriptional profiles is enriched for a transcriptional program consistent with high EGFR pathway activity, we suspect it is the combined effect of lost SWI/SNF activity together with an increased EGFR transcriptional program that is responsible for resistance to cetuximab as opposed to just an increased EGFR expression and/or increased downstream pathway activation. We have added these citations to the discussion and stressed the necessity to further investigate the molecular characterization of the interaction between the ARID1A deficiency, loss of SWI/SNF activity and the EGFR pathway in driving resistance to cetuximab in ARID1A mutant eRAS/BRAF WT tumors, which is beyond the scope of this manuscript.